# Sequential activation of transcriptional repressors promotes progenitor commitment by silencing stem cell identity genes

Noemi Rives-Quinto[1], Hideyuki Komori[1], Cyrina M Ostgaard[1,2], Derek H Janssens[1†], Shu Kondo[3], Qi Dai[4], Adrian W Moore[5], Cheng-Yu Lee[1,2,6]*

[1]Life Sciences Institute, University of Michigan, Ann Arbor, United States; [2]Department of Cell and Developmental Biology, University of Michigan Medical School, Ann Arbor, United States; [3]Invertebrate Genetics Laboratory, National Institute of Genetics, Mishima, Japan; [4]Department of Molecular Biosciences, The Wenner-Gren Institute, Stockholm University, Stockholm, Sweden; [5]Riken Center for Brain Science, Wako, Japan; [6]Division of Genetic Medicine, Department of Internal Medicine and Comprehensive Cancer Center, University of Michigan Medical School, Ann Arbor, United States

*For correspondence:
leecheng@umich.edu

Present address: †Basic Sciences Division, Fred Hutchinson Cancer Research Center, Seattle, United States

Competing interests: The authors declare that no competing interests exist.

**Abstract** Stem cells that indirectly generate differentiated cells through intermediate progenitors drives vertebrate brain evolution. Due to a lack of lineage information, how stem cell functionality, including the competency to generate intermediate progenitors, becomes extinguished during progenitor commitment remains unclear. Type II neuroblasts in fly larval brains divide asymmetrically to generate a neuroblast and a progeny that commits to an intermediate progenitor (INP) identity. We identified Tailless (Tll) as a master regulator of type II neuroblast functional identity, including the competency to generate INPs. Successive expression of transcriptional repressors functions through Hdac3 to silence *tll* during INP commitment. Reducing repressor activity allows re-activation of Notch in INPs to ectopically induce *tll* expression driving supernumerary neuroblast formation. Knocking-down *hdac3* function prevents downregulation of *tll* during INP commitment. We propose that continual inactivation of stem cell identity genes allows intermediate progenitors to stably commit to generating diverse differentiated cells during indirect neurogenesis.

## Introduction

Indirect generation of differentiated cell types through intermediate progenitors allows tissue-specific stem cells to scale their progeny output to accommodate the development of appropriately sized organs in proportion to organism sizes. Indirect neurogenesis by the outer subventricular zone (OSVZ) neural stem cells drives the evolution of lissencephalic brains to gyrencephalic brains in vertebrates (*Cárdenas and Borrell, 2020*). The competency to generate intermediate progenitors and the maintenance of an undifferentiated state must be coordinately downregulated in stem cell progeny to ensure generation of differentiated cell types. Polycomb-mediated gene repression is thought to inactivate stem-cell-specific genes during differentiation (*Tsuboi et al., 2019*). However, loss of function of Polycomb Repressive Complex 2 (PRC2) did not lead to intermediate progenitors reverting into neural stem cells (*Abdusselamoglu et al., 2019*). Thus, the mechanisms that extinguish stem cell functionality during progenitor commitment remains poorly understood.

Two distinct neuroblast lineages function together to generate the number of neurons requisite for the development of adult fly brains (*Farnsworth and Doe, 2017*; *Homem et al., 2015*; *Janssens and Lee, 2014*). Similar to ventricular zone neural stem cells, type I neuroblasts undergo direct neurogenesis. Type I neuroblasts repeatedly undergo asymmetric division to generate one daughter cell that remains a neuroblast and one sibling cell (ganglion mother cell [GMC]) that divides once to generate two neurons. By contrast, type II neuroblasts undergo indirect neurogenesis, similar to OSVZ neural stem cells. Type II neuroblasts continually undergo asymmetric division to self-renew and to generate a sibling cell that commits to an intermediate progenitor (INP) identity (*Bello et al., 2008*; *Boone and Doe, 2008*; *Bowman et al., 2008*). The Sp family transcription factor Buttonhead (Btd) and the ETS-1 transcription factor PointedP1 (PntP1) are specifically expressed in type II neuroblasts (*Komori et al., 2014b*; *Xie et al., 2014*; *Zhu et al., 2011*). Btd and PntP1 expression levels decrease as a newly generated immature INP transitions into a non-Asense-expressing (Ase$^-$) immature INP. Three to four hrs after this transition, an Ase$^-$ immature INP upregulates Ase expression as it progresses through INP commitment. Once INP commitment is complete, an Ase$^+$ immature INP transitions into an INP that undergoes six to eight rounds of asymmetric division. These molecularly defined intermediate stages during INP commitment provide critical landmarks to investigate the mechanisms that extinguish the activity of type II neuroblast functional identity genes.

Notch signaling maintains type II neuroblasts in an undifferentiated state, partially by poising transcription of the master regulator of differentiation *earmuff* (*erm*) (*Janssens et al., 2017*). During asymmetric neuroblast division, the TRIM-NHL protein Brain tumor (Brat) segregates into the newly generated immature INP and targets transcripts encoded by Notch downstream effector genes for RNA decay (*Bello et al., 2006*; *Betschinger et al., 2006*; *Komori et al., 2018*; *Lee et al., 2006b*; *Xiao et al., 2012*). *brat*-null brains accumulate thousands of supernumerary type II neuroblasts that originate from the reversion of newly generated immature INPs due to defects in the downregulation of Notch signaling (*Komori et al., 2014a*). In parallel to Brat, the nuclear protein Insensible (Insb) inhibits the activity of Notch downstream effector proteins during asymmetric neuroblast division, and Insb overexpression efficiently triggers premature differentiation in type II neuroblasts (*Komori et al., 2018*). This multi-layered gene control allows for the onset of Erm expression, coinciding with the termination of PntP1 expression. Erm expression is maintained in immature INPs but becomes downregulated in INPs (*Janssens et al., 2014*). Erm belongs to a family of transcription factors that are highly expressed in neural progenitors and can bind histone deacetylase 3 (Hdac3) (*Hirata et al., 2006*; *Koe et al., 2014*; *Levkowitz et al., 2003*; *Weng et al., 2010*). Erm prevents INP reversion by repressing gene transcription. In *erm*-null brains, INPs spontaneously revert to type II neuroblasts; however, this phenotype can be suppressed by knocking-down *Notch* function (*Weng et al., 2010*). Thus, Erm likely inactivates type II neuroblast functionality genes by promoting histone deacetylation.

By comparing mRNAs enriched in type II neuroblasts or immature INPs, we identified *tailless* (*tll*) as a master regulator of type II neuroblast functional identity. Tll is expressed in type II neuroblasts but not in INPs; moreover, Tll is necessary and sufficient for type II neuroblast functionality. Tll overexpression is sufficient to transform type I neuroblasts into type II neuroblasts, as indicated by changes in gene expression and an acquired competence for generating INPs. We identified *hamlet* (*ham*) as a new negative regulator of type II neuroblast maintenance. *ham* is expressed after *erm* in immature INP during INP commitment, and Erm and Ham function through Hdac3 to continually inactivate *tll*. Sequential inactivation during INP commitment suppresses *tll* activation by Notch signaling in INPs. We propose that silencing of the master regulator of stem cell functional identity allows intermediate progenitors to stably commit to the generation of differentiated cell types without reacquiring stem cell functionality.

## Results

### A novel transient over-expression strategy to identify regulators of type II neuroblast functional identity

Genes that regulate neuroblast functional identity should be expressed in type II neuroblasts and become rapidly downregulated in Ase$^-$ and Ase$^+$ immature INPs in wild-type brains (*Figure 1A*). To

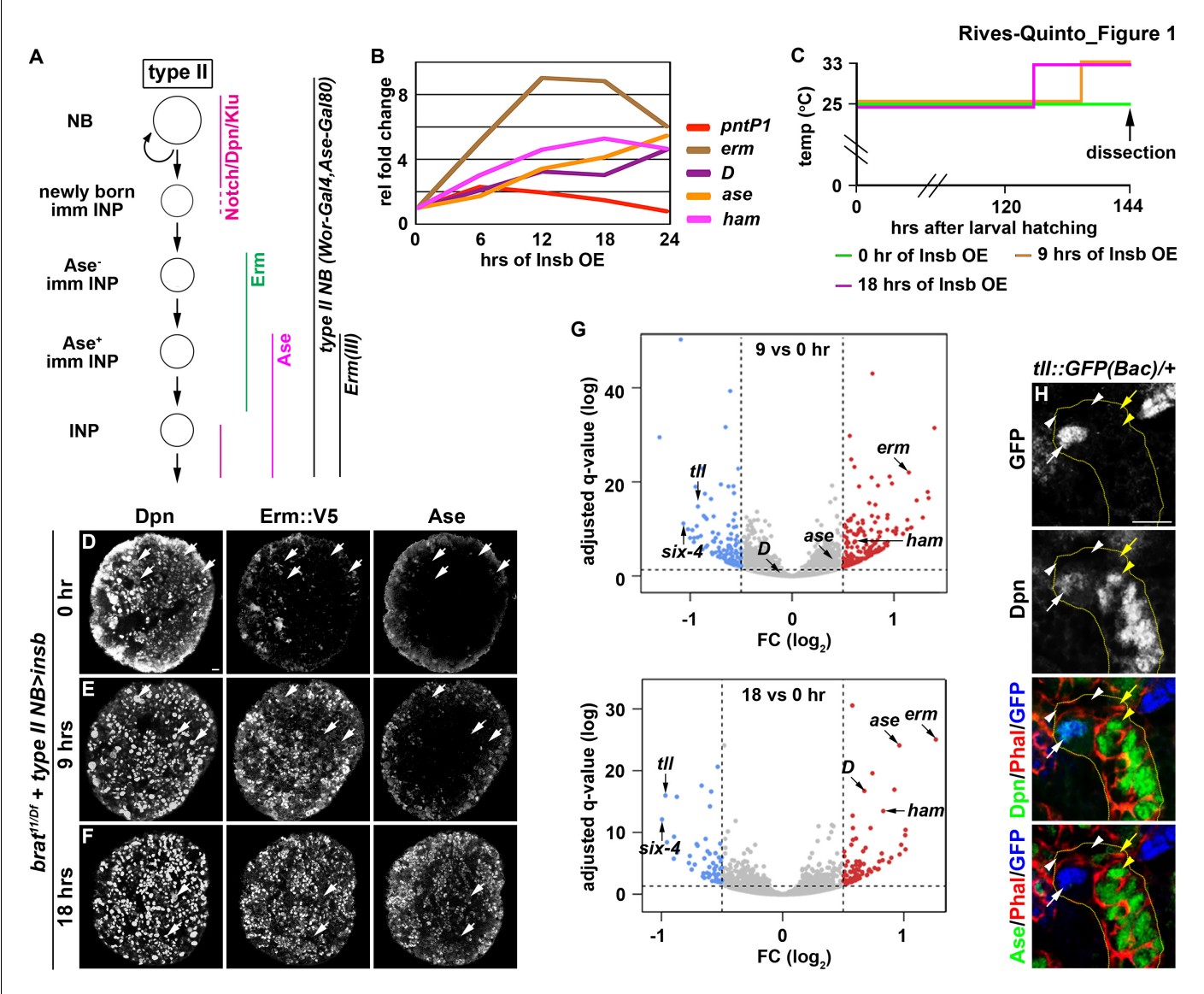

**Figure 1.** Identification of candidate regulators of type II neuroblast functional identity. (**A**) A diagram of the type II neuroblast lineage showing the expression patterns of genes and *Gal4* drivers used throughout this study. (**B**) Gene transcription profiles of *brat*-null brains transiently overexpressing Insb driven by a type II neuroblast Gal4. Supernumerary type II neuroblasts in *brat*-null brains transiently overexpressing Insb displayed patterns of gene transcription that are indicative of immature INPs undergoing INP commitment in wild-type brains. (**C**) A strategy to induce synchronized INP commitment in supernumerary type II neuroblasts in *brat*-null brains. Larvae carrying a *UAS-insb* transgene and a type II neuroblast Gal4 were collected and aged at 25°C. One third of larvae were shifted to 33°C at 126 or 135 hr after hatching to induce high levels of transient Insb expression, and the last one-third remained at 25°C serving as the source enriched for type II neuroblast-specific transcripts (time 0). (**D–F**) Images of *brat*-null brains transiently overexpressing Insb driven by a type II neuroblast Gal4 for 0, 9, or 18 hr. Transient overexpression of Insb first induced Erm and then Ase expression in supernumerary type II neuroblasts in *brat*-null brains. (**G**) Volcano plots showing fold-change of gene expression in *brat*-null brains transiently overexpressing Insb for 9 or 18 hr. (**H**) Tll expression pattern in the type II neuroblast lineage. The *tll::GFP(Bac)* transgene revealed the expression of endogenous Tll in type II neuroblasts but not in immature INPs and INPs. The following labeling applies to all images in this figure: yellow dashed line encircles a type II neuroblast lineage; white arrow: type II neuroblast; white arrowhead: Ase⁻ immature INP; yellow arrow: Ase⁺ immature INP; yellow arrowhead: INP. Scale bar, 10 μm.

The online version of this article includes the following source data and figure supplement(s) for figure 1:

**Figure supplement 1.** Time-course analysis of transient Insb overexpression in *brat*-null brains.

**Figure supplement 1—source data 1.** Quantification of total type II neuroblasts or INPs per *brat*-null brain lobe that transiently overexpressed Insb.

enrich for transcripts that fulfill these criteria, we tested if transient Insb overexpression induces supernumerary type II neuroblasts in *brat*-null brains to synchronously transition into INPs as in wild-type brains. The combination of *Wor-Gal4,AseGal80*, which is only active in type II neuroblasts, and *Tub-Gal80^{ts}*, which loses its inhibitory effect on Gal4 activity under non-permissive temperatures, allows for spatial and temporal control of Insb overexpression in all type II neuroblasts. We allowed larvae to grow at 25℃, and induced transient Insb overexpression for 6, 12, 18, or 24 hr by shifting the larvae to a non-permissive temperature of 33℃. Larvae that remained at 25℃ for the duration of this experiment served as the control (time 0). We assessed the effect of this transient over-expression strategy on cell identity by quantifying the total type II neuroblasts and INPs per brain lobe based on Deadpan (Dpn) and Ase expression (*Figure 1A*). Overexpressing Insb for 6 hr did not significantly affect the supernumerary neuroblast phenotype in *brat*-null brains (*Figure 1—figure supplement 1A*). By contrast, 12 or more hours of Insb overexpression led to a time-dependent decrease in supernumerary type II neuroblasts and a corresponding increase in INPs. These results demonstrate that transient Insb overexpression can induce supernumerary type II neuroblasts to synchronously transition into INPs in *brat*-null brains.

We next assessed if supernumerary type II neuroblasts transiently overexpressing Insb transition through identical intermediate stages to become INPs in *brat*-null brains as they do in wild-type brains (*Figure 1A*). We found that *pntP1* mRNA levels increased within the first 6 hr of Insb overexpression, and then continually declined (*Figure 1B*). By contrast, *erm* transcription rapidly increased in the first 12 hr of Insb overexpression, plateauing between 12 and 18 hr. *ase* transcription showed little change in the first 6 hr of Insb overexpression, and then steadily increased between 12 and 24 hr. The combined temporal patterns of *pntP1*, *erm,* and *ase* transcription strongly suggest that type II neuroblasts transiently overexpressing Insb for 6 hr in *brat*-null brains are at an equivalent stage as a newly generated immature INP transitioning to an Ase⁻ immature INP in wild-type brains. *brat*-null type II neuroblasts are at a stage equivalent to (1) Ase⁻ immature INPs following 6–12 hr of Insb overexpression and (2) Ase⁺ immature INPs following 12–24 hr of Insb overexpression. We examined the expression of endogenous Erm in *brat*-null type II neuroblasts transiently overexpressing Insb to confirm their identity. We generated an *erm::V5* allele by inserting a V5 epitope at the C-terminus of the *erm* reading frame, and showed that Erm::V5 recapitulates the published endogenous Erm expression pattern that we previously described (*Figure 1—figure supplement 1B*; *Janssens et al., 2014*). Indeed, following 9 hr of Insb overexpression most type II neuroblasts in *brat*-null brains expressed cell identity markers indicative of Ase⁻ immature INPs (Erm::V5⁺Ase⁻), whereas after 18 hr of Insb overexpression they expressed markers indicative of Ase⁺ immature INPs (Erm::V5⁺Ase⁺) (*Figure 1C–F*). These data indicate that supernumerary type II neuroblasts transiently overexpressing Insb indeed transition through identical intermediate stages during INP commitment in *brat*-null brains as in wild-type brains.

We predicted that a candidate regulator of type II neuroblast functional identity should become downregulated in *brat*-null brains following 9 and 18 hr of Insb overexpression. By sequencing mRNA in triplicate following the transient over-expression strategy (*Figure 1C*), we identified 76 genes that were reproducibly downregulated by 1.5-fold or more in *brat*-null brains overexpressing Insb for 9 hr (*Figure 1G*). Of these genes, *tll* was the most downregulated; similarly, *tll* was the most downregulated gene in *brat*-null brains overexpressing Insb for 18 hr. We validated the *tll* expression pattern in the type II neuroblast lineage using a bacterial artificial chromosome (BAC) transgene [*tll::GFP(Bac)*] where green fluorescent protein (GFP) is fused in frame with the *tll* reading frame. Consistent with previous study (*Bayraktar and Doe, 2013*), we detected Tll::GFP in type II neuroblasts but not in immature or mature INPs (*Figure 1H*). Tll::GFP expression is also detected in type I neuroblasts in the brain and in the ventral nerve cord, but at a significantly lower level than in type II neuroblasts (*Figure 1—figure supplement 1C–E*). Thus, *tll* is an excellent candidate for regulating type II neuroblast functionality.

## *tll* is a master regulator of type II neuroblast functional identity

We defined the type II neuroblast functional identity as the maintenance of an undifferentiated state and the competency to generate INPs. We first tested whether *tll* is required for maintaining type II neuroblasts in an undifferentiated state by overexpressing a *UAS-tll^{RNAi}* transgene in the type II neuroblast lineage under the control of the *Wor-Gal4,Ase-Gal80* driver (*Figure 2A*). Whereas control brains always contained eight type II neuroblasts per lobe, brains with *tll* function knocked down

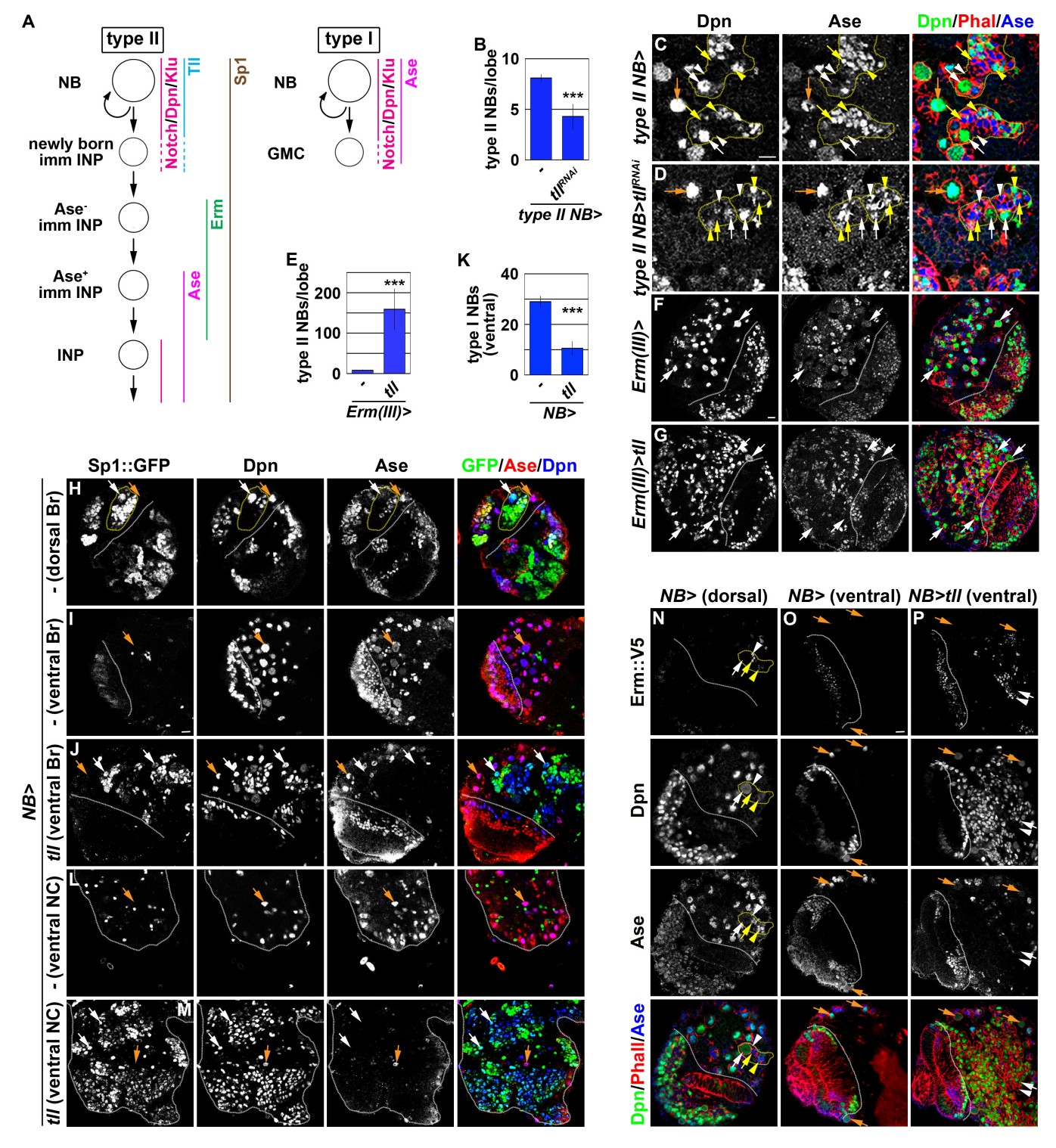

**Figure 2.** *tll* is necessary and sufficient for type II neuroblast functional identity. (**A**) A diagram showing the expression patterns of genes in the type I and II neuroblast lineages. (**B**) Quantification of total type II neuroblasts per brain lobe that overexpressed a *UAS-tll*<sup>RNAi</sup> transgene driven by a type II neuroblast Gal4. Knocking-down *tll* function reduced the number of type II neuroblasts from 8 to 4 per brain lobe. (**C–D**) Images of brains that overexpressed a *UAS-tll*<sup>RNAi</sup> transgene driven by a type II neuroblast Gal4. Knocking-down *tll* function led to premature differentiation in type II neuroblast as indicated by reduced cell diameter and ectopic Ase expression but did not affect the maintenance of type I neuroblasts. (**E**) Quantification of total type II neuroblasts per brain that overexpressed a *UAS-tll* transgene driven by an INP Gal4. *tll* overexpression in INPs led to

*Figure 2 continued on next page*

*Figure 2 continued*

supernumerary type II neuroblast formation. (F–G) Images of brains that overexpressed a *UAS–tll* transgene driven by an INP Gal4. (H–I) Images of *Sp1:: GFP(Bac)* brains. *Sp1::GFP* is detected in most cells in all type II neuroblast lineages that are exclusively located in the dorsal brain region, and is detected in few neurons in type I neuroblast lineages in the ventral brain region. (J) Images of the ventral region of *Sp1::GFP(Bac)* brains that overexpressed a *UAS-tll* transgene driven by a pan-neuroblast Gal4 (*Wor-Gal4*). Tll overexpression transforms type I neuroblasts (Ase$^+$Sp1::GFP$^-$) in the ventral brain region into type II neuroblasts (Ase$^-$Sp1::GFP$^+$). (K) Quantification of total ventral type I neuroblasts per brain lobe that overexpressed a *UAS-tll* transgene driven by a pan-neuroblast Gal4. Tll overexpression transforms 66% of type I neuroblasts in the ventral brain region into type II neuroblasts. (L–M) Images of the thoracic segments on the ventral nerve cord of *Sp1::GFP(Bac)* larvae that overexpressed a *UAS-tll* transgene driven by a pan-neuroblast Gal4 (*Wor-Gal4*). Tll overexpression transforms type I neuroblasts in the thoracic segments into type II neuroblasts. (N–O) Images of dorsal and ventral regions of *erm::V5* brains. Erm::V5 is detected in immature INPs in type II neuroblast lineages but is undetectable in type I neuroblast lineages in the ventral region of larval brain. (P) Images of *erm::V5* brains that overexpressed a *UAS-tll* transgene driven by a pan-neuroblast Gal4. Tll overexpression induced type I neuroblasts in the ventral-medial region of the brain to generate supernumerary type II neuroblasts interspersed with Erm::V5$^+$ immature INPs. The following labeling applies to all images in this figure: white dashed line separates the optic lobe from the brain; yellow dashed line encircles a type II neuroblast lineage; white arrow: type II neuroblast; white arrowhead: Ase$^-$ immature INP; yellow arrow: Ase$^+$ immature INP; yellow arrowhead: INP; orange arrow: type I neuroblast. Br: brain. NC: nerve cord. Scale bar, 10 μm. Bar graphs are represented as mean ± standard deviation. p-values: ***<0.005.

The online version of this article includes the following source data for figure 2:

**Source data 1.** Quantification of total type II neuroblasts per brain lobe that overexpressed a *UAS-tll$^{RNAi}$* transgene.
**Source data 2.** Quantification of total type II neuroblasts per brain that overexpressed a *UAS-tll* transgene.
**Source data 3.** Quantification of total ventral type I neuroblasts per brain lobe that overexpressed a *UAS-tll* transgene.

contained 4.3 ± 1.2 type II neuroblasts per lobe (n = 10 brains per genotype) (*Figure 2B*). A closer examination revealed that the remaining *tll* mutant neuroblasts showed reduced cell diameters and ectopically expressed Ase, two characteristics typically associated with INPs in the type II neuroblast lineage (*Figure 2A, C and D*). This result indicated that *tll* is required for maintaining type II neuroblasts in an undifferentiated state. We next tested whether *tll* is sufficient to re-establish a type II neuroblast-like undifferentiated state in INPs by misexpressing a *UAS-tll* transgene under the control of *Erm-Gal4(III)* (*Figure 1A*). Brains with *tll* misexpressed in INPs contained 150 ± 50 type II neuroblasts per lobe, whereas control brains always contained eight type II neuroblasts per lobe (n = 10 brains per genotype) (*Figure 2E–G*). Thus, Tll misexpression is sufficient to revert INPs to type II neuroblasts. These data together led us to conclude that *tll* is necessary and sufficient for maintaining type II neuroblasts in an undifferentiated state.

As *tll* is required to maintain type II neuroblasts in an undifferentiated state, direct assessment of its role in regulating the competency to generate INPs is not possible. As an alternative, we tested whether Tll overexpression is sufficient to induce ectopic type II neuroblast formation in the ventral brain region and in the ventral nerve cord that exclusively consist of type I neuroblasts. To unambiguously distinguish the two types of neuroblasts, we searched for robust protein markers of type II neuroblasts and their progeny. As *Sp1* mRNA is uniquely detected in type II neuroblasts (*Yang et al., 2016*), we evaluated Sp1 protein expression using an *Sp1::GFP(Bac)* transgene. We found that Sp1::GFP marks type II neuroblasts and most of their progeny that are found exclusively in the dorsal brain region (*Figure 2H*). Sp1::GFP is detected in some neurons but never in type I neuroblasts in the ventral brain region and in the ventral nerve cord (*Figure 2I and L*). Thus, Sp1::GFP is a new marker for the type II neuroblast lineage (*Figure 2A*). We overexpressed a *UAS-tll* transgene under the control of a pan-neuroblast driver (*Wor-Gal4*) in larval brains carrying the *Sp1::GFP(Bac)* transgene. While the ventral region of control brains contains 29 ± 2.1 type I neuroblasts, the same region of the brains overexpressing Tll contains 10.6 ± 2.5 type I neuroblasts and hundreds of ectopic type II neuroblasts (n = 10 brains per genotype) (*Figure 2I–K*). This result strongly suggests that Tll overexpression transforms greater than 60% of type I neuroblasts in the ventral brain region into type II neuroblasts. Similarly, Tll overexpression also induced hundreds of supernumerary type II neuroblasts in the ventral nerve cord (n = 10 brains per genotype) (*Figure 2L and M*). Thus, ectopic *tll* expression is sufficient to molecularly transform type I neuroblasts into type II neuroblasts.

Next, we tested if type I neuroblasts overexpressing *tll* can generate immature INPs that are marked by Erm::V5 expression (*Figure 2A and N*). We reproducibly observed Erm::V5$^+$ immature INPs in the ventral-medial region of the brain overexpressing Tll, but not in the ventral-lateral brain region (*Figure 2O and P*). This result strongly suggests that progeny of transformed type II

neuroblasts in the ventral-medial brain region assume an immature INP identity and then revert into supernumerary type II neuroblasts. By contrast, progeny of transformed type II neuroblasts in the ventral-lateral brain region appeared to rapidly revert into supernumerary neuroblasts instead of assuming an immature INP identity. Thus, *tll* overexpression is sufficient to molecularly and functionally transform type I neuroblasts into type II neuroblasts, and that *tll* is a master regulator of type II neuroblast functional identity.

## Ham is a new regulator of INP commitment

A previous study found that Suppressor of Hairless, the DNA-binding partner of Notch, binds the *tll* locus in larval brain neuroblasts, suggesting that *tll* is a putative Notch target (*Zacharioudaki et al., 2016*). Thus, mechanisms likely exist to silence *tll* during INP commitment, preventing re-activation of Notch signaling in INPs from ectopically inducing Tll expression and driving supernumerary type II neuroblast formation (*Figure 2A*). We predicted that genes required for silencing *tll* during INP commitment should become upregulated in *brat*-null brains following 9 and 18 hr of Insb overexpression, and mutations in these genes should lead to supernumerary type II neuroblast formation due to ectopic *tll* expression in INPs. From our RNA sequencing dataset, we identified genes that encode transcription factors and are upregulated in *brat*-null brains overexpressing Insb for 9 or 18 hr. These transcription factors include Erm, Dichaete (D), Ase, and Hamlet (Ham) (*Figure 1G*). Reverse transcriptase PCR confirmed that the transcript levels of these genes indeed become upregulated in *brat*-null brains overexpressing Insb (*Figure 1B*). Thus, these genes are candidates for inactivating *tll* during INP commitment.

We tested if any of the candidate genes is required for inactivating *tll* expression during INP commitment by knocking-down their function in *erm* hypomorphic (*erm^hypo*) brains. *erm^hypo* brains display a low frequency of INP reversion to type II neuroblasts, and provides a sensitized genetic background for identifying genes required to prevent INP reversion to type II neuroblasts (*Janssens et al., 2017*). Because ectopic *tll* expression in INPs leads to supernumerary type II neuroblast formation, reducing the function of genes that inactivate *tll* should enhance the supernumerary neuroblast phenotype in *erm^hypo* brains. We found that *erm^hypo* brains alone contained 31.5 ± 6.8 type II neuroblasts per lobe (n = 10 brains) (*Figure 3A*). Although reducing the function of most candidate genes did not enhance the supernumerary neuroblast phenotype in *erm^hypo* brains, knocking-down *ham* function with two different *UAS-RNAi* transgenes reproducibly enhanced the phenotype (n = 10 brains per transgene) (*Figure 3A*). Consistent with the effect of reducing *ham* function by RNAi, the heterozygosity of a deficiency that deletes the entire *ham* locus also enhanced the supernumerary neuroblast phenotype in *erm^hypo* brains (data not presented). Thus, *ham* likely plays a role in preventing INPs from reverting to supernumerary type II neuroblasts.

*ham* is the fly homolog of the vertebrate *Prdm16* gene that encodes a transcription factor and has been shown to play a key role in regulating cell fate decisions in multiple stem cell lineages (*Baizabal et al., 2018*; *Harms et al., 2015*; *Moore et al., 2002*; *Shimada et al., 2017*). A previous study concluded that *ham* regulates INP proliferation but does not play a role in suppressing supernumerary type II neuroblast formation (*Eroglu et al., 2014*). The discrepancy between our findings and the published observations might be due to the *ham^1* allele used in the previous study, which encodes a nearly full-length protein and exhibits similar protein stability as wild-type Ham (*Figure 3B*; *Moore et al., 2002*). We took two approaches to determine whether *ham* is required for preventing supernumerary type II neuroblast formation. First, we examined *ham* deficiency heterozygous brains and reproducibly observed a mild supernumerary type II neuroblast phenotype (9 ± 0.9 per lobe, n = 12 brains) (*Figure 3—figure supplement 1A*). Second, we knocked down *ham* function by overexpressing a *UAS-RNAi* transgene. We found that knocking down *ham* function also led to a mild but statistically significant increase in type II neuroblasts per lobe (9.8 ± 1.8; n = 20 brains) as compared to control brains (8 ± 0; n = 10 brains) (*Figure 3—figure supplement 1B*). To confirm that *ham* is indeed required for suppressing supernumerary type II neuroblast formation, we generated two new *ham* alleles, *ham^SK1* and *ham^SK4*, by CRISPR-Cas9 (*Figure 3B*). We characterized the effect of the *ham^SK1* or *ham^SK4* allele on Ham protein expression using a specific antibody. We found that *ham^SK1* or *ham^SK4* homozygous brains show undetectable Ham protein expression (*Figure 3C and D*; data not presented). This result strongly suggests that mutant Ham protein encoded by the *ham^SK1* or *ham^SK4* allele is unstable, and that these two new *ham* alleles are strongly loss-of-function mutants. We then tested if Ham protein is required for suppressing supernumerary

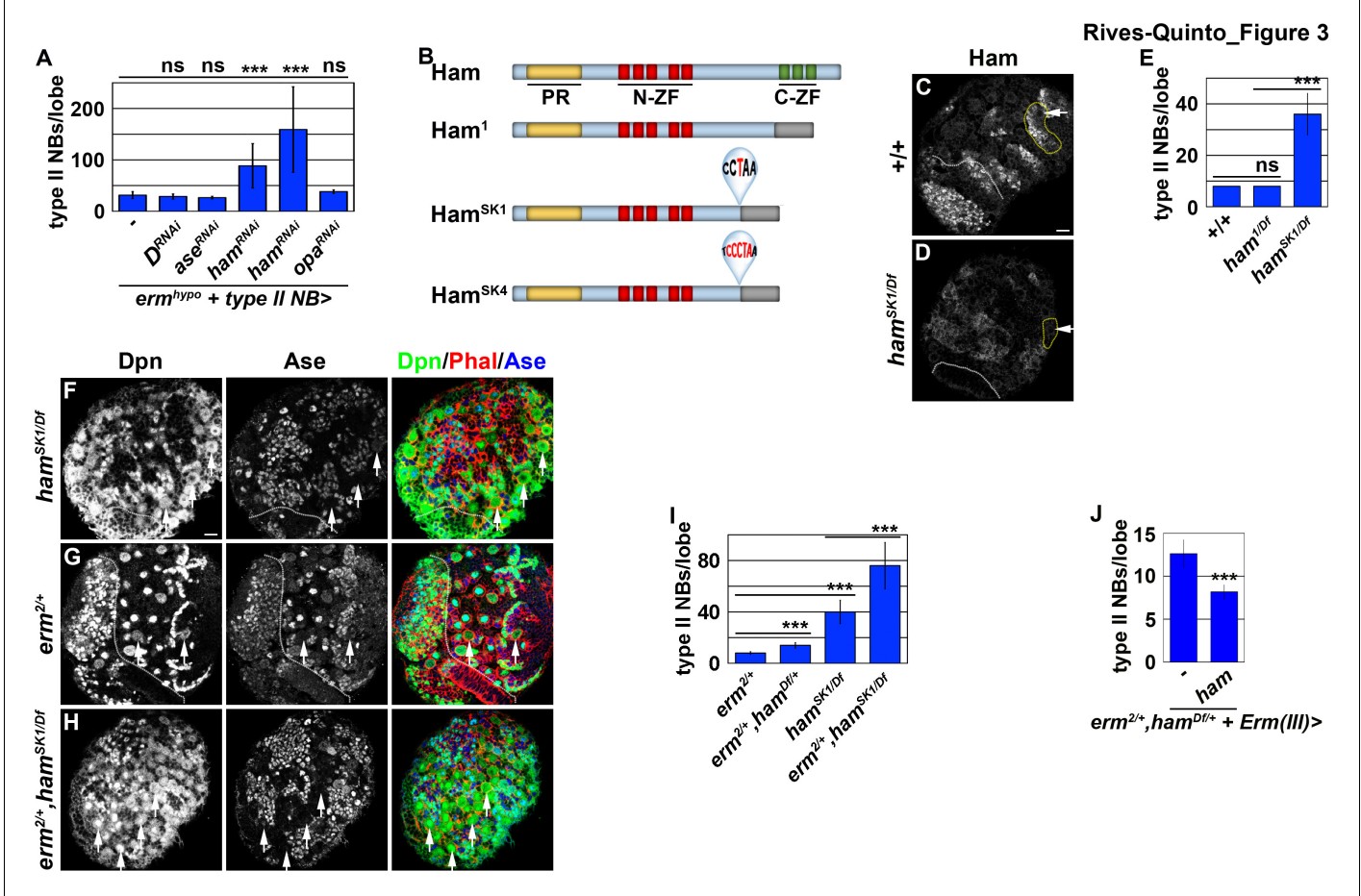

**Figure 3.** Ham is a novel regulator of INP commitment. (**A**) Quantification of total type II neuroblasts per *erm^hypo* brain lobe that overexpressed a *UAS-RNAi* transgene driven by a type II neuroblast Gal4. Knocking-down *ham* function consistently enhanced the supernumerary type II neuroblast phenotype in *erm^hypo* brains. (**B**) A diagram summarizing the lesions in *ham* alleles. The molecular lesions in *ham^SK1* and *ham^SK4* alleles were not independently verified. (**C–D**) Ham expression in wild-type or *ham^SK1* homozygous brains. Ham was detected in immature INPs and INPs in wild-type brains, but was undetectable in *ham^SK1* homozygous brains. (**E**) Quantification of total type II neuroblasts per *ham*-mutant brain lobe. *ham^SK1* homozygous but not *ham^1* homozygous brains displayed a supernumerary type II neuroblast phenotype. (**F–H**) Images of *ham* single mutant or *ham, erm* double mutant brains. The heterozygosity of *erm* alone had no effect on type II neuroblasts, but enhanced the supernumerary type II neuroblast phenotype in *ham^SK1* homozygous brains. (**I**) Quantification of total type II neuroblasts per brain lobe of the indicated genotypes. *erm* and *ham* function synergistically to prevent supernumerary type II neuroblast formation. (**J**) Quantification of total type II neuroblasts per *erm,ham* double heterozygous brain lobe that overexpressed a *UAS-ham* transgene driven by an INP Gal4. Overexpressing Ham in INPs rescued the supernumerary type II neuroblast phenotype in *erm,ham* double heterozygous brains. The following labeling applies to all images in this figure: white dashed line separates the optic lobe from the brain; yellow dashed line encircles a type II neuroblast lineage; white arrow: type II neuroblast. Scale bar, 10 μm. Bar graphs are represented as mean ± standard deviation. p-values: ***<0.005. ns: not significant.

The online version of this article includes the following source data and figure supplement(s) for figure 3:

**Source data 1.** Quantification of total type II neuroblasts per *erm^hypo* brain lobe that overexpressed a *UAS-RNAi* transgene.
**Source data 2.** Quantification of total type II neuroblasts per *ham*-mutant brain lobe.
**Source data 3.** Quantification of total type II neuroblasts per brain lobe of the indicated genotypes.
**Source data 4.** Quantification of total type II neuroblasts per *erm,ham* double heterozygous brain lobe that overexpressed a *UAS-ham* transgene.
**Figure supplement 1.** Ham functions synergistically with Erm to suppress supernumerary type II neuroblast formation.
**Figure supplement 1—source data 1.** Quantification of total type II neuroblasts per brain lobe that overexpressed a *UAS-ham^RNAi* transgene.

type II neuroblast formation. Indeed, *ham^SK1* homozygous brains contained 40 ± 10 type II neuro-blasts per lobe (n = 15 brains) (*Figure 3E and F*). Because Ham is detectable in Ase+ immature INPs and remains expressed in all INPs (*Eroglu et al., 2014*), we conclude that *ham* is a novel regulator of INP commitment.

*ham* knock-down drastically enhanced the supernumerary neuroblast phenotype in *erm*<sup>hypo</sup> brains, leading us to hypothesize that *ham* functions together with *erm* to suppress INPs from reverting to supernumerary type II neuroblasts. We tested this hypothesis by examining a potential genetic interaction between *erm* and *ham* during INP commitment. Although the heterozygosity of *erm* did not increase total type II neuroblasts (8.1 ± 0.3 per lobe; n = 11 brains), it enhanced the supernumerary neuroblast phenotype in *ham* deficiency heterozygous brains and in *ham*<sup>SK1</sup> homozygous brains (14 ± 2.2 vs. 76.2 ± 18.3 type II neuroblasts per lobe; n = 14 brains per genotype) (*Figure 3G–I*). We conclude that Ham functions synergistically with Erm to suppress supernumerary type II neuroblast formation.

## Ham promotes stable INP commitment by repressing gene transcription

Re-activation of Notch signaling in INPs drives supernumerary type II neuroblast formation in *erm*-null brains (*Weng et al., 2010*). Therefore, we hypothesized that supernumerary type II neuroblasts in *ham*-null brains might also originate from INPs. Whereas *erm,ham* heterozygous brains alone contained 12.7 ± 1.6 type II neuroblasts per lobe (n = 9 brains), overexpressing *ham* in INPs driven by *Erm-Gal4(III)* rescued the supernumerary neuroblast phenotype in *erm,ham* double heterozygous brains (8.2 ± 0.8 type II neuroblasts per lobe; n = 24 brains) (*Figure 3J*). This result strongly suggests that supernumerary type II neuroblasts originate from INPs in *ham*-null brains. We generated GFP-marked mosaic clones derived from single type II neuroblasts to confirm the origin of supernumerary type II neuroblasts in *ham*-null brains. In wild-type clones (n = 9 clones), the parental type II neuroblast was always surrounded by Ase<sup>-</sup> and Ase<sup>+</sup> immature INPs (*Figure 4A*). By contrast, supernumerary neuroblasts in *ham*<sup>SK1</sup> homozygous clones (2.5 ± 1.7 type II neuroblasts per clone; n = 11 clones) were always located far away from parental neuroblasts and surrounded by Ase<sup>+</sup> cells that were most likely Ase<sup>+</sup> immature INPs and ganglion mother cells (*Figure 4B and C*). It is highly unlikely that supernumerary neuroblasts in *ham*<sup>SK1</sup> homozygous clones originated from symmetric neuroblast division based on their location relative to parental neuroblasts and the cell types that surround them. Thus, we conclude that Ham suppresses INP reversion to supernumerary type II neuroblasts.

Because Erm suppresses supernumerary type II neuroblast formation by repressing gene transcription, Ham likely prevents INP reversion as a transcriptional repressor. We reasoned that Ham might function through its N-terminal zinc finger to prevent INP reversion based on our finding that *ham*<sup>SK1</sup> but not *ham*<sup>1</sup> homozygous clones displayed a supernumerary neuroblast phenotype (*Figure 4C*). Consistent with this hypothesis, mis-expressing Ham<sup>ΔC-ZF</sup> triggered premature differentiation in type II neuroblasts, identical to mis-expression of full-length Ham (n = 10 brains per genotype) (*Figure 4E*). We tested if overexpressing Ham<sup>ΔC-ZF</sup> in INPs can rescue the supernumerary neuroblast phenotype in *ham*<sup>SK1</sup> homozygous brains by incubating larvae at 33°C for 96 hr. Incubating *ham*<sup>SK1</sup> homozygous larvae at an elevated temperature leads to a severer supernumerary neuroblast phenotype than at 25°C (*Figures 3E* and *4F*). Overexpressing Ham<sup>ΔC-ZF</sup> rescued the supernumerary neuroblast phenotype in *ham*<sup>SK1</sup> homozygous brains to a similar extent as overexpressing full-length Ham (11.1 ± 1.4 vs. 11.9 ± 1.3 type II neuroblasts per lobe; n = 11 or 9 brains, respectively) (*Figure 4F*). Under identical conditions, overexpressing a constitutive transcriptional repressor form of Ham containing only the N-terminal zinc fingers (ERD::Ham<sup>N-ZF</sup>) also rescued the supernumerary neuroblast phenotype in *ham*<sup>SK1</sup> homozygous brains (14 ± 2.3 type II neuroblasts per lobe; n = 9 brains) (*Figure 4D and F*). These results indicate that Ham functions through the N-terminal zinc finger to repress the transcription of genes that can trigger INP reversion to supernumerary type II neuroblasts.

Finally, we tested if the N-terminal zinc finger of Ham mediates target gene recognition. We mis-expressed a constitutive transcriptional activator form of Ham containing only the N-terminal zinc-finger motif (VP16::Ham<sup>N-ZF</sup>) (*Figure 4D*). We found that VP16::Ham<sup>N-ZF</sup> misexpression in type II neuroblasts was sufficient to induce supernumerary neuroblast formation (17.1 ± 4.7 type II neuroblasts per lobe; n = 10 brains) (*Figure 4G*). Because VP16::Ham<sup>N-ZF</sup> can exert a dominant-negative effect, we conclude that Ham suppresses INP reversion by recognizing target genes through its N-terminal zinc-finger motif and repressing their transcription.

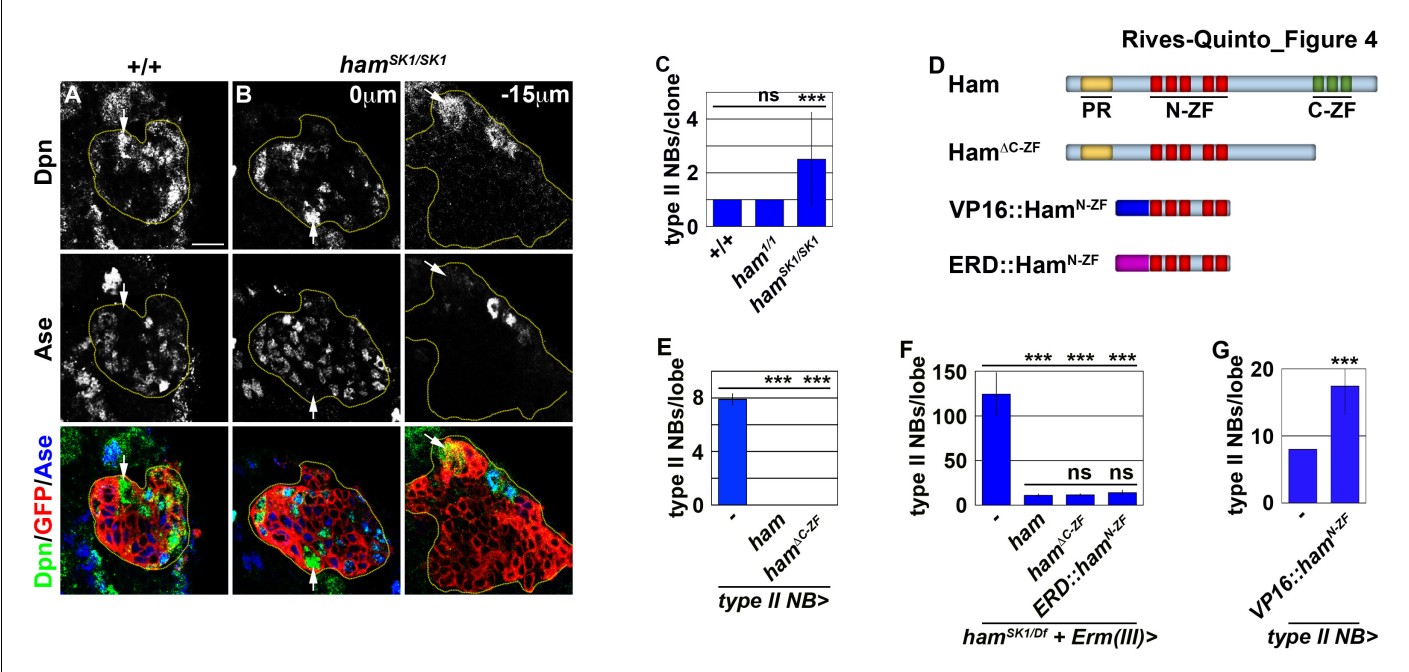

**Figure 4.** Ham suppresses INP reversion by repressing gene transcription. (**A–B**) Images of wild-type or *ham*[SK1] homozygous type II neuroblast mosaic clones. Supernumerary neuroblasts (−15 µm) in *ham*[SK1] homozygous clones were always located far from the parental neuroblast (0 µm), and were surrounded by Ase[+] cells. (**C**) Quantification of total neuroblasts per *ham*[1] or *ham*[SK1] homozygous type II neuroblast clone. *ham*[SK1] homozygous clones contained supernumerary neuroblasts but *ham*[1] homozygous clones did not. (**D**) A diagram summarizing *UAS-ham* transgenes used in this study. (**E**) Quantification of total type II neuroblasts per brain lobe that overexpressed a *UAS-ham* transgene driven by a type II neuroblast Gal4. Overexpressing full-length Ham or Ham[ΔC-ZF] led to premature differentiation in type II neuroblasts. (**F**) Quantification of total type II neuroblasts per *ham*[SK1] homozygous brain lobe that overexpressed various *UAS-ham* transgenes driven by an INP Gal4. Overexpressed full-length Ham, Ham[ΔC-ZF] or ERD:: Ham[N-ZF] rescued the supernumerary neuroblast phenotype in *ham*[SK1] homozygous brains. (**G**) Quantification of total type II neuroblasts per brain lobe that overexpressed a *UAS-VP16::ham*[N-ZF] transgene driven by a type II neuroblast Gal4. Overexpressing VP16::Ham[N-ZF] led to supernumerary type II neuroblast formation. The following labeling applies to all images in this figure: yellow dashed line encircles a type II neuroblast lineage; white arrow: type II neuroblast. Scale bar, 10 µm. Bar graphs are represented as mean ± standard deviation. Ppvalues: \*\*\*<0.005. ns: not significant.

The online version of this article includes the following source data for figure 4:

**Source data 1.** Quantification of total neuroblasts per *ham*[1] or *ham*[SK1] homozygous type II neuroblast clone.
**Source data 2.** Quantification of total type II neuroblasts per brain lobe that overexpressed a *UAS-ham* transgene.
**Source data 3.** Quantification of total type II neuroblasts per *ham*[SK1] homozygous brain lobe that overexpressed various *UAS-ham* transgenes.
**Source data 4.** Quantification of total type II neuroblasts per brain lobe that overexpressed a *UAS-VP16::ham*[N-ZF] transgene.

## Inactivation during INP commitment relinquishes Notch's ability to activate *tll* in INPs

Our findings strongly suggest that Erm and Ham inactivate *tll* during INP commitment, preventing the re-activation of Notch signaling from triggering *tll* expression in INPs (*Figure 5A*). Therefore, we tested if Erm and Ham indeed inactivate *tll* by reducing *tll* function in *erm,ham* double heterozygous brains. *tll* heterozygous brains contained 8.2 ± 0.4 type II neuroblasts per lobe, whereas *erm,ham* double heterozygous brains contained 11 ± 1.2 type II neuroblasts per lobe (n = 10 or 25 brains, respectively) (*Figure 5B*). The heterozygosity of *tll* consistently suppressed the supernumerary neuroblast phenotype in *erm,ham* double heterozygous brains (8.4 ± 0.6 type II neuroblasts per lobe; n = 22 brains) (*Figure 5B*). This result strongly supports our hypothesis that Erm and Ham inactivate *tll* during INP commitment. Consistent with this interpretation, we found that Tll::GFP becomes ectopically expressed in INPs in *erm,ham* double heterozygous brains (*Figure 5C*). In these brains, we reproducibly observed small cells expressing Tll::GFP and Deadpan (Dpn) but not Ase that were most likely supernumerary type II neuroblasts newly derived from INP reversion (*Figure 5D*). Furthermore, we found that one copy of the *tll::GFP(BAC)* transgene mildly enhanced the supernumerary

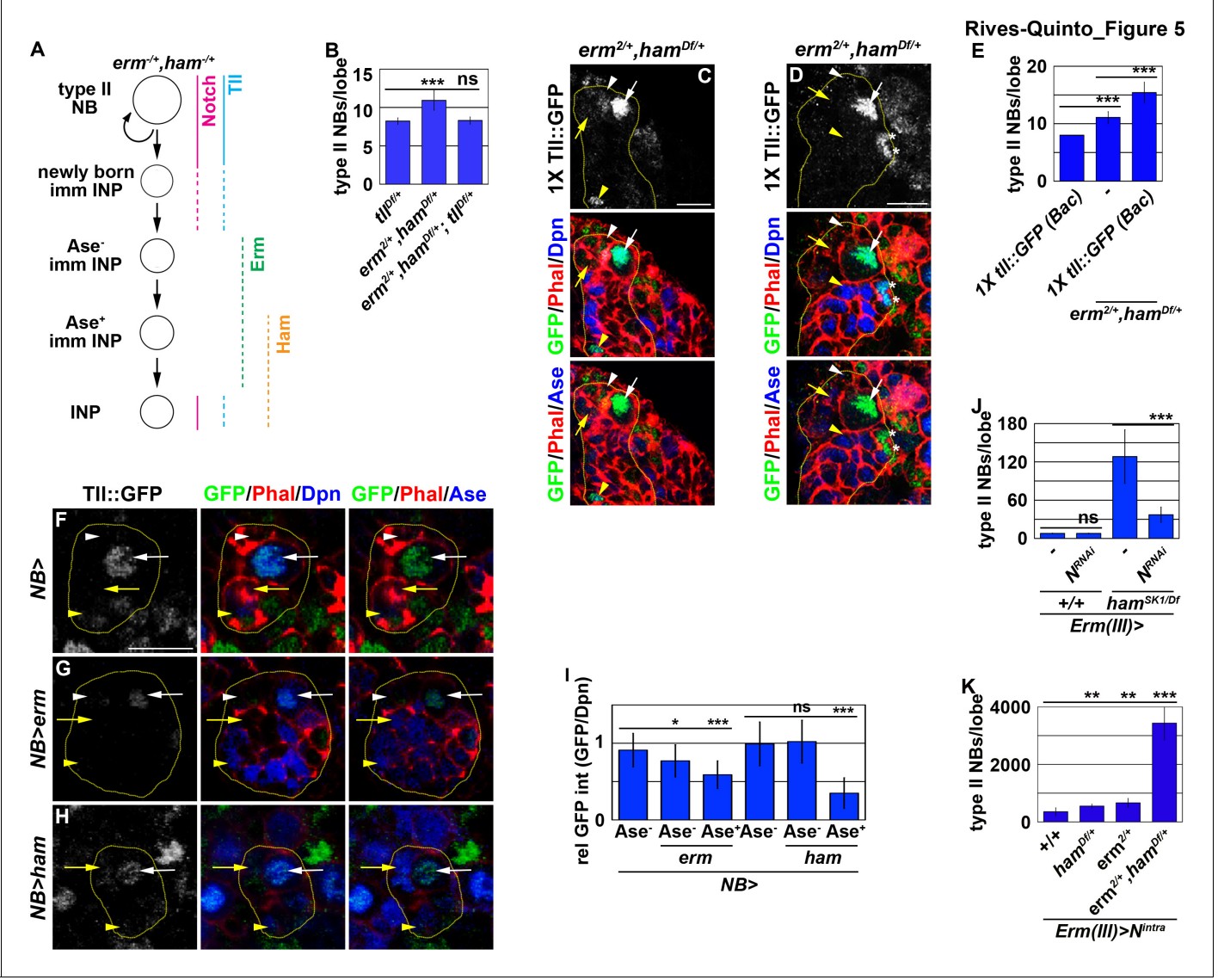

**Figure 5.** Erm- and Ham-mediated repression renders *tll* refractory for activation by Notch. (**A**) A diagram depicting our hypothesis that ectopic activation of *tll* in INPs leads to supernumerary type II neuroblasts in *erm,ham* double heterozygous brains. (**B**) Quantification of total type II neuroblasts per brain lobe that was *erm,ham* double heterozygous or *erm,ham,tll* triple heterozygous. Heterozygosity of *tll* suppressed the supernumerary neuroblast phenotype in *erm,ham* double heterozygous brains. (**C–D**) Images of *erm,ham* double heterozygous brains that carries a *tll::GFP(BAC)* transgene. Tll::GFP becomes ectopically expressed in INPs and supernumerary type II neuroblasts (*) in *erm,ham* double heterozygous brains. (**E**) Quantification of total type II neuroblasts *erm,ham* double heterozygous brain lobe that carried one copy of the *tll::GFP(BAC)* transgene. One copy of the *tll::GFP(BAC)* transgene was sufficient to enhance the supernumerary type II neuroblast phenotype in *erm,ham* double heterozygous brains. (**F–H**) Images of type II neuroblasts that mis-expressed a *UAS-erm* or *UAS-ham* transgene driven by a pan-neuroblast Gal4. Erm or Ham mis-expression drastically reduced Tll::GFP expression in type II neuroblasts. (**I**) Quantification of Tll::GFP expression relative to Dpn expression in type II neuroblasts that mis-expressed a *UAS-erm* or *UAS-ham* transgene driven by a pan-neuroblast Gal4. Erm mis-expression reduced Tll::GFP expression in type II neuroblasts before the onset of Ase expression, whereas Ham mis-expression reduced Tll::GFP expression in type II neuroblasts after the onset of Ase expression. (**J**) Quantification of total type II neuroblasts per wild-type or *ham^{SK1}* homozygous brain lobe that overexpressed a *UAS-N^{RNAi}* transgene driven by an INP Gal4. Knocking-down *Notch* function in INPs suppressed the supernumerary type II neuroblast phenotype in *ham^{SK1}* homozygous brains. (**K**) Quantification of total type II neuroblasts per *erm* or *ham* heterozygous brain lobe that overexpressed a *UAS-N^{intra}* transgene driven by an INP Gal4. Overexpressing *N^{intra}* in INPs more efficiently induced supernumerary neuroblasts in *erm,ham* double heterozygous brains than in *erm* or *ham* heterozygous brains. The following labeling is applicable to all panels of images in this figure: yellow dashed line encircles a type II neuroblast lineage; white arrow: type II neuroblast; white arrowhead: Ase⁻ immature INP; yellow arrow: Ase⁺ immature INP; yellow arrowhead: INP; *: supernumerary type II neuroblasts. Scale bar, 10 μm. Bar graphs are represented as mean ± standard deviation. p-values: **<0.05, ***<0.005. ns: not significant.

The online version of this article includes the following source data for figure 5:

**Source data 1.** Quantification of total type II neuroblasts per brain lobe that was *erm,ham* double heterozygous or *erm,ham,tll* triple heterozygous.
**Source data 2.** Quantification of total type II neuroblasts *erm,ham* double heterozygous brain lobe that carried one copy of the *tll::GFP(BAC)* transgene.

**Source data 3.** Quantification of Tll::GFP expression relative to Dpn expression in type II neuroblasts that mis-expressed a *UAS-erm* or *UAS-ham* transgene.

**Source data 4.** Quantification of total type II neuroblasts per wild-type or *ham^SK1* homozygous brain lobe that overexpressed a *UAS-N^RNAi* transgene.

**Source data 5.** Quantification of total type II neuroblasts per *erm* or *ham* heterozygous brain lobe that overexpressed a *UAS-N^intra* transgene.

neuroblast phenotype in *erm,ham* double heterozygous brains (15.4 ± 1.8 vs. 11.1 ± 1 type II neuroblasts per lobe; n = 12 or 11 brains, respectively) (*Figure 5E*). These data strongly support our model that Erm and Ham inactivate *tll* during INP commitment. We tested this model by overexpressing *erm* or *ham* in type II neuroblasts that carry one copy of the *tll::GFP(Bac)* transgene. We found that Tll::GFP expression in type II neuroblasts overexpressing Erm was mildly reduced prior to upregulation of Ase expression, and declined further following upregulation of Ase expression (*Figure 5F, G and I*). By contrast, Tll::GFP expression in type II neuroblasts overexpressing Ham was unaffected prior to upregulation of Ase expression, but rapidly declined following upregulation of Ase expression (*Figure 5H and I*). Together, these data strongly support our model that sequential inactivation by Erm and Ham during INP commitment renders *tll* refractory to activation in INPs.

Because *tll* is a putative target of Notch, we tested if inactivation of *tll* by Erm and Ham relinquishes the ability of Notch signaling to activate *tll* in INPs. We knocked-down *Notch* function in INPs in wild-type or *ham^SK1* homozygous brains. Knock-down of *Notch* function in INPs had no effect on type II neuroblasts in wild-type brains (7.9 ± 0.3 type II neuroblasts per lobe; n = 17 brains) (*Figure 5J*). However, in *ham^SK1* homozygous brains, knocking-down *Notch* function in INPs reduced the number of type II neuroblasts per lobe from 128 ± 39.6 to 36.4 ± 11.6 (n = 9 brains per genotype) (*Figure 5J*). This result indicates that re-activation of Notch signaling triggers INP reversion to supernumerary type II neuroblasts in *ham*-null brains. We extended our analyses to test if inactivation by Erm and Ham reduces the competency to respond to Notch signaling. Consistently, overexpressing constitutively activated Notch (Notch^intra) induced a drastically more severe supernumerary neuroblast phenotype in *erm,ham* double heterozygous brains (3,437.6 ± 586.8 type II neuroblasts per lobe; n = 9 brains) than in *erm* or *ham* single heterozygous brains (687 ± 134.3 vs. 588.6 ± 77.6 type II neuroblasts per lobe; n = 9 or 11 brains, respectively) (*Figure 5K*). Thus, we propose that inactivation by Erm and Ham during INP commitment renders *tll* refractory to Notch signaling in INPs (*Figure 5A*).

## Inactivation by Hdac3 relinquishes Notch's ability to activate *tll* in INPs

To prevent Notch signaling from activating *tll* in INPs, Erm and Ham must function through chromatin-modifying proteins. We first tested if Erm and Ham inactivate *tll* by promoting sequential chromatin changes during INP commitment. We overexpressed full-length Ham or Ham^ΔC-ZF driven by *Erm-Gal4(III)* in *erm*-null brains. *erm*-null brains alone contained 1824.6 ± 520.7 type II neuroblasts per lobe (n = 13 brains) (*Figure 6A* and *Figure 6—figure supplement 1A*). Overexpressing full-length Ham or Ham^ΔC-ZF strongly suppressed the supernumerary neuroblast phenotype in *erm*-null brains (34.5 ± 8.3 vs. 64.6 ± 18.3 neuroblasts per lobe; n = 13 or 7 brains, respectively) (*Figure 6A* and *Figure 6—figure supplement 1B*). This result indicates that Ham over-expression can replace endogenous Erm function to suppress INP reversion, suggesting that these two transcriptional repressors function through an identical chromatin-modifying protein to inactivate *tll* during INP commitment.

We predicted that decreasing the activity of a chromatin-modifying protein required for Ham-mediated gene inactivation during INP commitment should enhance the supernumerary neuroblast phenotype in *ham* heterozygous brains. We knocked-down the function of genes known to contribute to the inactivation of gene transcription in *ham* heterozygous brains. Reducing the activity of PRC2 or heterochromatin protein 1, as well as decreasing the recruitment of Heterochromatin Protein 1, had no effect on the supernumerary neuroblast phenotype in *ham* heterozygous brains (*Figure 6B*). While reducing the activity of nucleosome remodelers alone led to a mild supernumerary type II neuroblast phenotype, it did not further enhance the supernumerary neuroblast phenotype in *ham* heterozygous brains (*Figure 6C*). By contrast, reducing the activity of histone deacetylase 3 (Hdac3) alone resulted in a mild supernumerary type II neuroblast phenotype and significantly enhanced the supernumerary neuroblast phenotype in *ham* heterozygous brains (*Figure 6C*). These results led us to conclude that Ham likely functions through Hdac3 to prevent

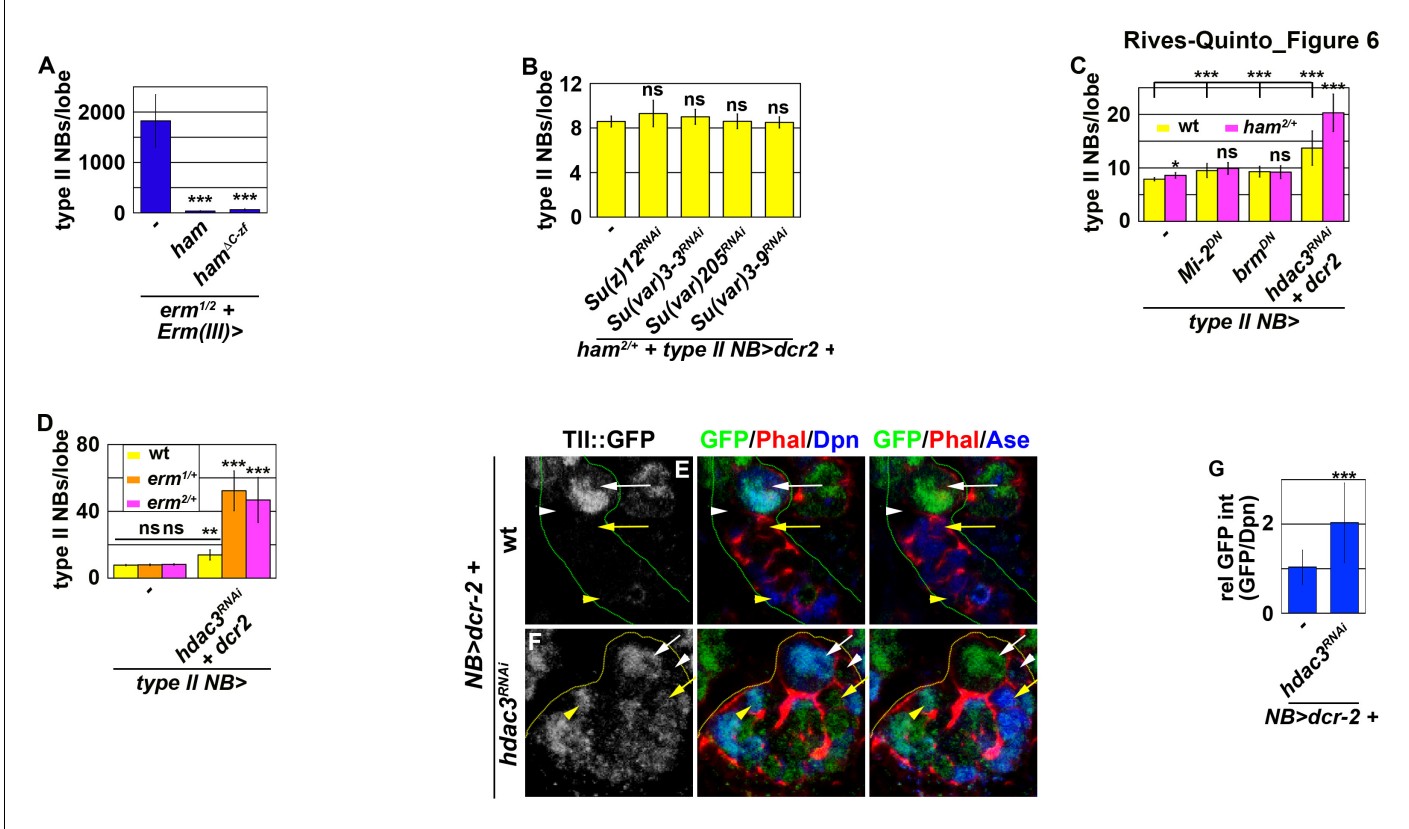

**Figure 6.** Erm and Ham function through Hdac3 to prevent INPs from reverting to type II neuroblasts. (**A**) Quantification of total type II neuroblasts per brain lobe that overexpressed a *UAS-ham* transgene driven by an INP Gal4. Overexpressing full-length Ham or Ham$^{\Delta C\text{-}ZF}$ in INPs suppressed the supernumerary type II neuroblast phenotype in *erm*-null brains. (**B**) Quantification of total type II neuroblasts per *ham* heterozygous brain lobe that overexpressed a *UAS-RNAi* transgene driven by a type II neuroblast Gal4. Reducing activity of the indicated chromatin complex did not increase INP reversion into supernumerary type II neuroblasts in *ham* heterozygous brains. (**C**) Quantification of total type II neuroblasts per *ham* heterozygous brain lobe that overexpressed a *UAS* transgene driven by a type II neuroblast Gal4. Reducing Hdac3 activity increased INP reversion into supernumerary type II neuroblasts in *ham* heterozygous brains. (**D**) Quantification of total type II neuroblasts per *erm* heterozygous brain lobe that overexpressed a *UAS-hdac3$^{RNAi}$* transgene driven by a type II neuroblast Gal4. Reducing Hdac3 activity increased INP reversion into supernumerary type II neuroblasts in *erm* heterozygous brains. (**E–F**) Images of *tll::GFP(Bac)* brains that overexpressed a *UAS-hdac3$^{RNAi}$* transgene driven by a pan-neuroblast Gal4. Reducing Hdac3 activity in type II neuroblasts led to ectopic Tll::GFP expression in immature INPs and INPs. (**G**) Quantification of Tll::GFP expression relative to Dpn expression in INPs derived from type II neuroblasts that overexpressed a *UAS-hdac3$^{RNAi}$* transgene. Reducing Hdac3 activity in type II neuroblasts led to ectopic Tll::GFP expression in INPs. The following labeling is applicable to all panels of images in this figure: yellow dashed line encircles a type II neuroblast lineage; white arrow: type II neuroblast; white arrowhead: Ase⁻ immature INP; yellow arrow: Ase⁺ immature INP; yellow arrowhead: INP. Bar graphs are represented as mean ± standard deviation. p-values: **<0.05, ***<0.005. ns: not significant.

The online version of this article includes the following source data and figure supplement(s) for figure 6:

**Source data 1.** Quantification of total type II neuroblasts per brain lobe that overexpressed a *UAS-ham* transgene.

**Source data 2.** Quantification of total type II neuroblasts per *ham* heterozygous brain lobe that overexpressed various *UAS* transgenes.

**Source data 3.** Quantification of total type II neuroblasts per *erm* heterozygous brain lobe that overexpressed a *UAS-hdac3$^{RNAi}$* transgene.

**Source data 4.** Quantification of Tll::GFP expression relative to Dpn expression in INPs derived from type II neuroblasts that overexpressed a *UAS-hdac3$^{RNAi}$* transgene.

**Figure supplement 1.** Ham overexpression in Ase⁺ immature INPs suppressed INP reversion in *erm*-null brains.

INP reversion. We next tested if Erm also functions through Hdac3 to prevent INP reversion. *erm* heterozygous brains alone did not display a supernumerary type II neuroblast phenotype (*Figure 6D*). Knocking-down *hdac3* function in *erm* heterozygous brains led to a more than three-fold increase in supernumerary type II neuroblasts as compared to wild-type brains (*Figure 6D*). These data strongly support a model that Hdac3 functions together with Erm and Ham to silence *tll* during INP commitment.

We tested if *hdac3* is indeed required for silencing *tll* during INP commitment by overexpressing a *UAS-hdac3^{RNAi}* transgene in type II neuroblasts. Consistent with our hypothesis, knock down of *hdac3* function in type II neuroblasts and their progeny led to ectopic Tll::GFP expression in immature INPs and INPs (*Figure 6E–G*). These results confirmed that Hdac3 is required for inactivating *tll* during INP commitment. Thus, we conclude that Erm and Ham likely function through Hdac3 to prevent INP from reverting to supernumerary type II neuroblasts by continually inactivating *tll*.

## Discussion

The expansion of OSVZ neural stem cells, which indirectly produce neurons by initially generating intermediate progenitors, drives the evolution of lissencephalic brains to gyrencephalic brains (*Cárdenas and Borrell, 2020*; *Delaunay et al., 2017*; *Di Lullo and Kriegstein, 2017*). Recent studies have revealed important insights into genes and cell biological changes that lead to the formation of OSVZ neural stem cells (*Fujita et al., 2020*; *Namba et al., 2020*). However, the mechanisms controlling the functional identity of OSVZ neural stem cells, including the competency to generate intermediate progenitors, remain unknown. In this study, we have provided compelling evidence demonstrating that Tll is necessary and sufficient for the maintenance of an undifferentiated state and the competency to generate intermediate progenitors in type II neuroblasts (*Figure 7*). We also showed that two sequentially activated transcriptional repressors, Erm and Ham, likely function through Hdac3 to silence *tll* during INP commitment ensuring normal indirect neurogenesis in larval brains. We propose that continual inactivation of stem cell functional identity genes by histone deacetylation allows intermediate progenitors to stably commit to generating sufficient and diverse differentiated cells during neurogenesis.

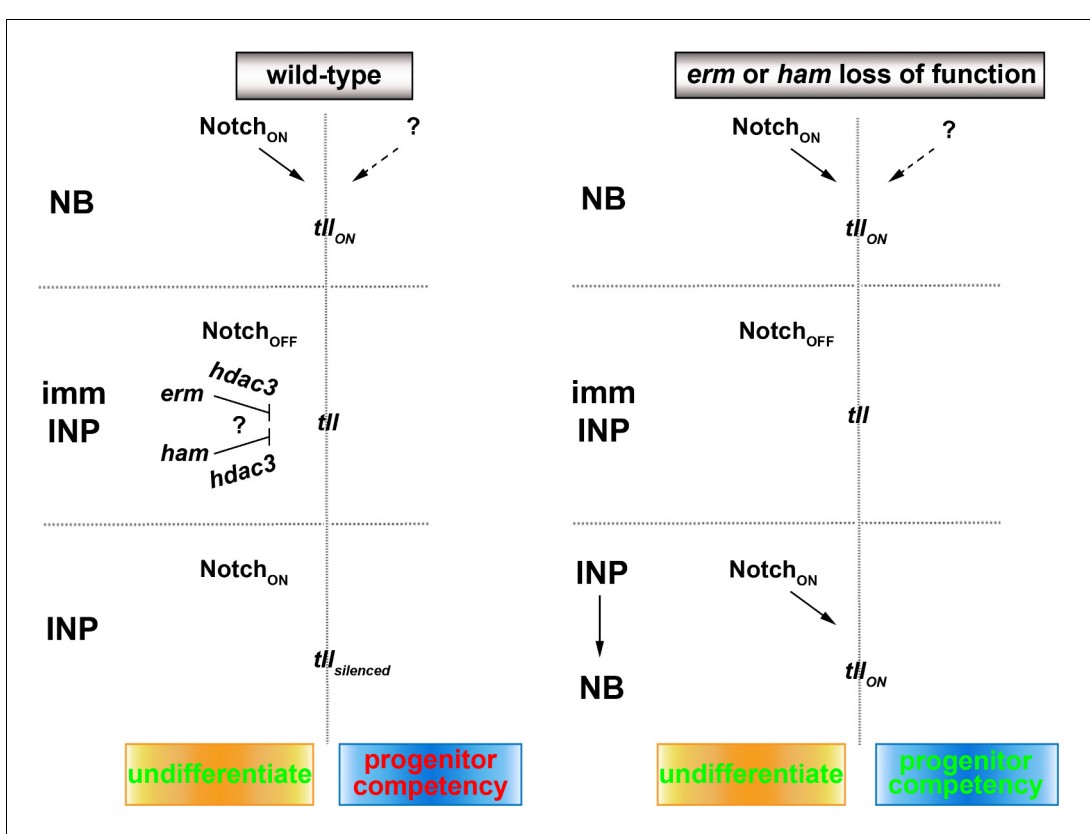

**Figure 7.** A proposed model for the regulation of type II neuroblast functionality. We propose that Erm and Ham recruits Hdac3 to silence *tll* during INP commitment, preventing re-activation of Notch signaling in INPs from triggering Tll expression in wild-type brains. The *tll* locus remains in an activatable state in *erm*- or *ham*-null brains, and re-activation of Notch signaling in INPs triggers aberrant Tll expression driving INP reversion to supernumerary type II neuroblasts.

# Regulation of the maintenance of an undifferentiated state and the competency to generate intermediate progenitors in neural stem cells

Stem cell functional identities encompass the maintenance of an undifferentiated state and other unique functional features, such as the competency to generate intermediate progenitors. Because genetic manipulation of Notch signaling perturbs the regulation of differentiation during asymmetric stem cell division (*Imayoshi et al., 2010*; *Kageyama et al., 2008*), the role of Notch in regulating other stem cell functions remains poorly understood. In the fly type II neuroblast lineage, overexpressing Notch[intra] or the downstream transcriptional repressors Dpn, E(spl)mγ, and Klumpfuss (Klu) induces the formation of supernumerary type II neuroblasts at the expense of generating immature INPs via the inhibition of *erm* activation (*Janssens et al., 2017*). These results indicate that the Notch-Dpn/E(spl)mγ/Klu axis provides an evolutionarily conserved mechanism to maintain neural stem cells in an undifferentiated state. Although overexpressing Notch[intra] but not Dpn, E(spl)mγ, and Klu in combination in INPs is sufficient to drive supernumerary type II neuroblast formation, Notch[intra] overexpression is not sufficient to transform a type I neuroblast into a type II neuroblast (data not presented). Thus, we propose that Notch functions as a general activator of genes expression in type II neuroblasts, and specific regulators of type II neuroblast functional identities must exist.

Our study strongly suggests that *tll* functions as a master regulator of type II neuroblast functional identities (*Figure 7*). Identical to *Notch*, *tll* is necessary for maintaining type II neuroblasts in an undifferentiated state and is sufficient to induce INP reversion into type II neuroblasts (*Figure 2B–G*; *Hakes and Brand, 2020*). Uniquely, high levels of Tll is sufficient to molecularly transform greater than 60% of type I neuroblasts in the ventral brain region into type II neuroblasts (*Figure 2H–M*). Brain regionalization leads to distinct degrees of sensitivity to Tll overexpression. For example, type I neuroblasts in the dorsal-anterior region of the brain are resistant to Tll-induced lineage transformation. By contrast, Tll overexpression can transform most, if not all, type I neuroblasts in the ventral-lateral and ventral-medial regions of the brain into type II neuroblasts. High levels of Tll expression in ventral-lateral type I neuroblasts leads to accumulation of mostly supernumerary type II neuroblasts and very few Erm::V5[+] immature INPs. This result phenocopies Tll overexpression driven by strong drivers in the type II neuroblast lineages, and suggests that neuroblast progeny rapidly reacquire a neuroblast identity instead of an immature INP identity (data not presented) (*Hakes and Brand, 2020*). Tll overexpression in ventral-medial type I neuroblasts leads to the formation of supernumerary type II neuroblasts interspersed with Erm::V5[+] immature INPs, mimicking Tll overexpression driven by moderate drivers in the type II neuroblast lineages. We speculate that progeny of transformed type II neuroblasts in the ventral-medial region of the brain can assume an immature INP identity and then revert into supernumerary type II neuroblasts. These data strongly support a model that Tll is a potent activator of type II neuroblast functional identities.

A key question regarding the proposed function of Tll in regulating type II neuroblast functional identities is how it mechanistically links to genes previously shown to control these characteristics. ChIP-seq on fly embryonic nuclear extract using the Tll::GFP(Bac) transgenic protein identified hundreds of putative Tll target genes that include all previously characterized regulators of type II neuroblast functional identities (*Davis et al., 2018*). Tll binds *dpn*, *E(spl)mγ* and *klu* loci in embryos, suggesting that Tll likely regulates their expression. The phenotypic effects of loss- and gain-of-function of *tll* on type II neuroblasts mimic those of *dpn*, *E(spl)mγ* and *klu*. The vertebrate homolog of Tll, Tlx, has been shown to function as a transcriptional activator during neurogenesis (*Sun et al., 2017*). Thus, Tll might maintain type II neuroblasts in an undifferentiated state by promoting *dpn*, *E(spl)mγ* and *klu* expression. Tll also binds *pntP1* and *btd* loci in embryos. Similar to Tll overexpression, mis-expression of PntP1 or Btd can transform type I neuroblasts in the ventral brain region into type II neuroblasts (*Komori et al., 2014b*; *Xie et al., 2014*; *Zhu et al., 2011*). Thus, it is plausible that *tll* functions through *pntP1* or *btd* to regulate the competency to generate INPs in type II neuroblasts. These results strongly support our model that Tll is key component of the regulatory mechanism that endows type II neuroblasts with lineage-specific functional identities. Future experiments to validate the mechanistic links between tll and genes that regulate various lineage-specific functional characteristics will allow for the establishment of gene regulatory circuits that regulate type II neuroblast functional identities.

## Successive transcriptional repressor activity inactivates stem cell functional identity genes during progenitor commitment

The identification of *ham* as a putative regulator of INP commitment was unexpected given that a previously published study concluded that Ham functions to limit INP proliferation (*Eroglu et al., 2014*). Ham is the fly homolog of Prdm16 in vertebrates and has been shown to play a key role in regulating cell fate decisions in multiple stem cell lineages (*Baizabal et al., 2018*; *Harms et al., 2015*; *Moore et al., 2002*; *Shimada et al., 2017*). Prdm16 contains two separately defined zinc-finger motifs, with each likely recognizing unique target genes. Prdm16 can also function through a variety of cofactors to activate or repress target gene expression, independent of its DNA-binding capacity. Thus, Ham can potentially inactivate stem cell functionality genes via one of several mechanisms. By using a combination of previously isolated alleles and new protein-null alleles, we demonstrated that the N-terminal zinc-finger motif is required for Ham function in immature INPs. Based on the overexpression of a series of chimeric proteins containing the N-terminal zinc-finger motif, our data indicate that Ham prevents INP reversion to supernumerary type II neuroblasts by recognizing target genes via the N-terminal zinc-finger motif and possibly repressing their transcription. Our results suggest that Ham prevents INPs from reverting to supernumerary type II neuroblasts by possibly repressing target gene transcription.

A key question raised by our study is why two transcriptional repressors that seemingly function in a redundant manner are required to prevent INPs from reverting to supernumerary type II neuroblasts. INP commitment lasts approximately 6–8 hr following the generation of an immature INP (*Berger et al., 2012*; *Homem et al., 2014*); after this time, the immature INP transitions into an INP. *erm* is poised for activation in type II neuroblasts and becomes rapidly activated in the newly generated immature INP less than 90 min after its generation (*Janssens et al., 2017*). As such, Erm-mediated transcriptional repression allows for the rapid inactivation of type II neuroblast functional identity genes. Because Erm expression rapidly declines in INPs when Notch signaling becomes reactivated, a second transcriptional repressor that becomes activated after Erm and whose expression is maintained throughout the life of an INP is required to continually inactivate type II neuroblast functional identity genes. Ham is an excellent candidate because it becomes expressed in immature INPs 3–4 hr after the onset of Erm expression and is detected in all INPs. Similar to Erm, Ham recognizes target genes and represses their transcription. Furthermore, *ham* functions synergistically with *erm* to prevent INP reversion to supernumerary type II neuroblasts, and overexpressed Ham can partially substitute for endogenous Erm. Thus, Erm- and Ham-mediated transcriptional repression renders type II neuroblast functional identity genes refractory to activation by Notch signaling throughout the lifespan of the INP, ensuring the generation of differentiated cell types rather than supernumerary type II neuroblasts instead.

## Sustained inactivation of stem cell functional identity genes distinguishes intermediate progenitors from stem cells

Genes that specify stem cell functional identity become refractory to activation during differentiation, but the mechanisms that restrict their expression are poorly understood due to a lack of lineage information. Researchers have proposed several epigenetic regulator complexes that may restrict neural stem-cell-specific gene expression in neurons (*Hirabayashi and Gotoh, 2010*; *Ronan et al., 2013*). We knocked-down the function of genes that were implicated in restricting neural stem cell gene expression during differentiation in order to identify chromatin regulators that are required to inactivate type II neuroblast functional identity genes during INP commitment. Surprisingly, we found that only Hdac3 is required for both Erm- and Ham-mediated suppression of INP reversion to type II neuroblasts. Our finding is consistent with a recent study showing that blocking apoptosis in lineage clones derived from PRC2-mutant type II neuroblasts did not lead to supernumerary neuroblast formation (*Abdusselamoglu et al., 2019*). Our data strongly suggest that genes specifying type II neuroblast functional identity, such as *tll*, are likely silenced rather than decommissioned in INPs. This result is supported by the finding that overexpressing Notch[intra] but not Notch downstream transcriptional repressors in INPs can re-establish a type II neuroblast-like undifferentiated state. We speculate that continual histone deacetylation is required to counter the transcriptional activator activity of endogenous Notch and silence *tll* in INPs (*Figure 7*). By contrast, the chromatin in the *tll* locus might be close and inaccessible to the Notch transcriptional activator complex in type

I neuroblasts; thus, overexpressing Notch[intra] cannot transform type I neuroblasts into type II neuroblasts. A key remaining question is what transcription factor is required to maintain the chromatin in the *tll* loci in an open state. Insights into regulation of the competency of the *tll* locus to respond to activated Notch signaling might improve our understanding of the molecular determinants of OSVZ neural stem cells.

# Materials and methods

## Fly genetics and transgenes

Fly crosses were carried out in 6-oz plastic bottles, and eggs were collected on apple caps in 8 hr intervals. Newly hatched larvae were genotyped and allowed to grow on corn meal caps. Larvae were shifted to 33°C for 72 hr io induce *UAS*-transgene expression for overexpression or knock down studies prior to dissection.

Larvae for MARCM analyses were genotyped at hatching and allowed to grow at 25°C for 24 hr. Larvae were then shifted to a 37°C water bath for 90 minto induce clones. Heat-shocked larvae were allowed to recover and grow at 25°C for 72 hr prior to dissection.

Transgenes were inserted into the *pUAST-attB M{3xP3-RFP.attP}ZH-86Fb* docking site using ΦC31 integrase-mediated transgenesis (*Bischof and Basler, 2008*). DNA injections were carried out by BestGene Inc or Genetivision Inc, and transgenic flies were identified in the F1 generation based on their red eye color.

## Immunofluorescent staining and antibodies

Larval brains were dissected in PBS and fixed in 100 mM PIPES (pH 6.9), 1 mM EGTA, 0.3% Triton X-100, and 1 mM $MgSO_4$ containing 4% formaldehyde for 23 min. Fixed brain samples were washed with PBST (1XPBS and 0.3% Triton X-100). After removing the fix solution, samples were incubated with primary antibodies for 3 hr at room temperature. Three hours later, samples were washed with PBST and then incubated with secondary antibodies overnight at 4°C. On the next day, samples were washed with PBST and then equilibrated in ProLong Gold antifade mountant (ThermoFisher Scientific). The confocal images were acquired on a Leica SP5 scanning confocal microscope (Leica Microsystems, Inc). More than 10 brains per genotype were used to obtain data in each experiment.

## RNA extraction and qRT-PCR

Total RNA was extracted from control or treated *brat[11/Df(2L)Exel8040]* larvae carrying *2xTub-Gal80[ts]*, *Wor-Gal4* and *UAS-insb* transgenes using TRIzol (ThermoFisher Scientific) and mRNA was purified using the RNeasy Micro Kit (Qiagen) according to the manufacturer's protocol. First strand cDNA was synthesized using a 1[st] Strand cDNA Synthesis Kit for RT-PCR [AMV] (Roche) according to the manufacturer's protocol. cDNA is amplified by using gene-specific primers. qRT-PCR was performed using ABsolute QPCR SYBR Green ROX Mix (ThermoFisher Sientific). Data were analyzed by comparative CT method, and the relative mRNA expression was presented.

## Sample preparation and RNA sequencing

Control or treated *brat[11/Df(2L)Exel8040]* larvae carrying *2xTub-Gal80[ts]*, *Wor-Gal4* and *UAS-insb* transgenes were cultured in corn meal caps. One third of the larvae (18 hr Insb OE sample) were allowed to grow at 25°C for 102 hr and then transferred to 33°C to induce transient Insb overexpression for 18 hr. One-third of the larvae (9 hr Insb OE sample) were allowed to grow at 25°C for 111 hr and then transferred to 33°C to induce transient Insb overexpression for 9 hr. The remaining third of the larvae (0 hr Insb OE) were allowed to grow at 25°C for 120 hr. Fifty brains from each group were isolated on the same day, and immediately processed for RNA extraction. Biological triplicate brain samples were collected. Paired-end Poly-A-mRNA libraries were generated to be assayed by Illumina HiSeq 4000. Differential genes expression was based on absolute linear fold change >or equal 1.5.

## Bioinformatics: RNA-seq analysis

Biological triplicate brain samples for each time point were sequenced using Illumina HiSeq 4000. The samples had around 46, 45 and 44 million reads. Quality of the raw reads for each sample was

checked using FastQC (http://www.bioinformatics.bbsrc.ac.uk/projects/fastqc). We align the reads to the dm6 reference genome including mRNAs (http://www.genome.ucsu.edu/) with HISAT2 (version 2.1.0) and default parameters. Alignment results were validated through a second round of FASTQ followed by quantification and analysis. For differential expression analysis, Stringtie (version 1.3.4) quantifies expression at the gene and isoforms levels and DESeq2 (version 1.20.0) performs differential expression testing. We compared transcriptional profile between 0 hr vs 9 hr, 0 hr vs 18 hr, and 9 hr vs 18 hr. We identified genes and transcripts as being differentially expressed based on absolute linear fold change >or equal 1.5.

## Quantification and statistical analysis

The Image J software was used to quantify the number of cells of interest. Dpn single-channel confocal images were used to assign the area the type II Neuroblasts or INP nucleus. All biological replicates were independently collected and processed. All statistical analyses were performed using a two-tailed Student's T-test, a p-value<0.05,<0.005, and<0.0005 were indicated by (*), (**) and (***), respectively in figures.

## Acknowledgements

We thank Drs. E Lai and J Knoblich for providing us with reagents. We thank the Advance Genomics Core and Bioinformatics Core for technical assistance. We thank Dr. Melissa Harrison, Ms. Elizabeth Lawson, and members of the Lee lab for helpful discussions, and Science Editors Network for editing the manuscript. We thank the Bloomington *Drosophila* Stock Center, FlyORF, and the Vienna *Drosophila* RNAi Center for fly stocks. We thank BestGene Inc and GenetiVision for generating the transgenic fly lines. This work is supported by NIH grants R01NS107496 and R01NS111647.

## Additional information

### Funding

| Funder | Grant reference number | Author |
| --- | --- | --- |
| National Institute of Neurological Disorders and Stroke | R01NS107496 | Hideyuki Komori Cheng-Yu Lee |
| National Institute of Neurological Disorders and Stroke | R01NS111647 | Hideyuki Komori Cheng-Yu Lee |

The funders had no role in study design, data collection and interpretation, or the decision to submit the work for publication.

### Author contributions

Noemi Rives-Quinto, Hideyuki Komori, Data curation, Formal analysis, Investigation, Methodology, Writing - original draft; Cyrina M Ostgaard, Data curation, Formal analysis, Investigation; Derek H Janssens, Data curation, Formal analysis, Investigation, Methodology; Shu Kondo, Formal analysis, Investigation, Methodology; Qi Dai, Methodology; Adrian W Moore, Cheng-Yu Lee, Conceptualization, Data curation, Formal analysis, Supervision, Funding acquisition, Investigation, Methodology, Writing - original draft, Project administration, Writing - review and editing

### Author ORCIDs

Hideyuki Komori https://orcid.org/0000-0001-9764-5654
Shu Kondo http://orcid.org/0000-0002-4625-8379
Cheng-Yu Lee https://orcid.org/0000-0003-2291-1297

### Decision letter and Author response

Decision letter https://doi.org/10.7554/eLife.56187.sa1
Author response https://doi.org/10.7554/eLife.56187.sa2

## Additional files

### Supplementary files

• Transparent reporting form

### Data availability

Sequencing data have been deposited in GEO under accession codes GSE152636.

The following dataset was generated:

| Author(s) | Year | Dataset title | Dataset URL | Database and Identifier |
|---|---|---|---|---|
| Rives-Quinto N, Komori H, Lee CY | 2020 | Sequential activation of transcriptional repressors promotes progenitor commitment by silencing stem cell identity genes | https://www.ncbi.nlm.nih.gov/geo/query/acc.cgi?acc=GSE152636 | NCBI Gene Expression Omnibus, GSE152636 |

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

# Appendix 1

**Appendix 1—key resources table**

| Reagent type (species) or resource | Designation | Source or reference | Identifiers | Additional information |
|---|---|---|---|---|
| Antibody | anti-GFP (Chicken polyclonal) | Aves Labs, INC. | Cat#GFP-1020, RRID:AB_2307313 | IF(1:2000) |
| Antibody | anti-V5 (Mouse monoclonal) | ThermoFisher Scientific | Cat#R960-25, RRID:AB_2556564 | IF(1:500) |
| Antibody | anti-Ase (Rabbit polyclonal) | *Weng et al., 2010* doi: 10.1016/j.devcel.2009.12.007. | | IF(1:400) |
| Antibody | anti-Hamlet (Rabbit polyclonal) | *Eroglu et al., 2014* doi: 10.1016/j.cell.2014.01.053. | | IF(1:50) |
| Antibody | anti-Dpn (Rat monoclonal) | *Lee et al., 2006a* doi: 10.1038/nature04299. | clone 11D1BC7.14 | IF(1:2) |
| Antibody | Alexa Fluor 488 AffiniPure Anti-Chicken IgY (IgG) (H+L) (Donkey polyclonal) | Jackson Immuno Research Laboratories, INC. | Cat#703-545-155, RRID:AB_2340375 | IF(1:500) |
| Antibody | Alexa Fluor 647 AffiniPure anti-Rat IgG (H+L) (Goat polyclonal) | Jackson Immuno Research Laboratories, INC. | Cat#112-605-167 RRID:AB_2338404 | IF(1:500) |
| Antibody | anti-Mouse IgG (H+L) Highly Cross-Adsorbed Secondary Antibody, Alexa Fluor 488 (Goat polyclonal) | ThermoFisher Scientific | Cat#A-11029, RRID:AB_2534088 | IF(1:500) |
| Antibody | anti-Rabbit IgG (H+L) Highly Cross-Adsorbed Secondary Antibody, Alexa Fluor 488 (Goat polyclonal) | ThermoFisher Scientific | Cat#A-11034, RRID:AB_2576217 | IF(1:500) |
| Antibody | anti-Rabbit IgG (H+L) Highly Cross-Adsorbed Secondary Antibody, Alexa Fluor 546 (Goat polyclonal) | ThermoFisher Scientific | Cat#A-11035, RRID:AB_2534093 | IF(1:500) |
| Other | Rhodamine Phalloidin | ThermoFisher Scientific | Cat#R415 | IF(1:100) |
| Genetic reagent (*D. melanogaster*) | *brat[11]/CyO, Actin-GFP* | *Lee et al., 2006b* doi: 10.1016/j.devcel.2006.01.017. | | |
| Genetic reagent (*D. melanogaster*) | *w[1118]; Df(2L) Exel8040/CyO* | Bloomington Drosophila Stock Center | BDSC: 7847 FlyBase: FBst0007847; RRID:BDSC_7847 | FlyBase symbol: *Df(2L)Exel8040/CyO* |

*Continued on next page*

*Appendix 1—key resources table continued*

| Reagent type (species) or resource | Designation | Source or reference | Identifiers | Additional information |
|---|---|---|---|---|
| Genetic reagent (*D. melanogaster*) | $y^1$, M{vas-int.Dm}ZH-2A $w^*$; M{UAS-insb$^{FL}$-myc} ZH-86Fb | *Komori et al., 2018* doi: 10.1101/gad.320333.118. | | |
| Genetic reagent (*D. melanogaster*) | Wor-Gal4(II) | *Lee et al., 2006a* doi: 10.1038/nature04299. | | |
| Genetic reagent (*D. melanogaster*) | Wor-Gal4(III) | *Weng et al., 2010* doi: 10.1016/j.devcel.2009.12.007. | | |
| Genetic reagent (*D. melanogaster*) | $y^1$, $w^*$; P{tubPGAL80}LL10, P{neoFRT}40A/CyO | Bloomington *Drosophila* Stock Center | BDSC: 5192 FlyBase: FBst0005192; RRID:BDSC_5192 | FlyBase symbol: $y^1$, $w^*$; P{tubPGAL80}LL10, P{neoFRT}40A/CyO |
| Genetic reagent (*D. melanogaster*) | P{hsFLP}$^1$, P{tubP-GAL80}LL1, $w^*$, P{neoFRT}19A; P{UAS-mCD8::GFP.L}LL5 | Bloomington *Drosophila* Stock Center | BDSC: 5134 FlyBase: FBst0005134; RRID:BDSC_5134_ | FlyBase symbol: P{hsFLP}$^1$, P{tubP-GAL80}LL1, $w^*$, P{neoFRT}19A; P{UAS-mCD8::GFP.L}LL5 |
| Genetic reagent (*D. melanogaster*) | $y^1$ $w^*$; PBac{y[+mDint2] w[+mC]=tll EGFP.S} VK00037 | Bloomington *Drosophila* Stock Center | BDSC: 30874 FlyBase: FBst0030874; RRID:BDSC_30874 | FlyBase symbol: $y^1$ $w^*$; PBac{y[+mDint2] w[+mC]=tll EGFP.S} VK00037 |
| Genetic reagent (*D. melanogaster*) | Erm-Gal4 (II) | *Pfeiffer et al., 2008* doi: 10.1073/pnas.0803697105. | | |
| Genetic reagent (*D. melanogaster*) | Erm-Gal4 (III) | *Pfeiffer et al., 2008* doi: 10.1073/pnas.0803697105. | | |
| Genetic reagent (*D. melanogaster*) | tll$^{RNAi}$: $y^1$ sc* $v^1$ sev$^{21}$; P{TRiP.HMS01316} attP2 | Bloomington *Drosophila* Stock Center | BDSC: 34329 FlyBase: FBst0034329; RRID:BDSC_34329 | FlyBase symbol: $y^1$ sc* $v^1$ sev$^{21}$; P{TRiP.HMS01316}attP2 |
| Genetic reagent (*D. melanogaster*) | M{UAS-tll.ORF-VN} ZH-86Fb | FlyORF | F004752 FBst0502964; RRID:FlyORF_ F004752 | FlyBase symbol: M{UAS-tll.ORF-VN}ZH-86Fb |
| Genetic reagent (*D. melanogaster*) | Ase-Gal80 (II) | *Neumüller et al., 2011* doi: 10.1016/j.stem.2011.02.022. | | |
| Genetic reagent (*D. melanogaster*) | erm$^1$/CyO, Act-GFP | *Weng et al., 2010* doi: 10.1016/j.devcel.2009.12.007. | | |
| Genetic reagent (*D. melanogaster*) | erm$^2$/CyO, Act-GFP | *Weng et al., 2010* doi: 10.1016/j.devcel.2009.12.007. | | |
| Genetic reagent (*D. melanogaster*) | UAS-erm | *Weng et al., 2010* doi: 10.1016/j.devcel.2009.12.007. | | |

*Continued on next page*

*Appendix 1—key resources table continued*

| Reagent type (species) or resource | Designation | Source or reference | Identifiers | Additional information |
|---|---|---|---|---|
| Genetic reagent (*D. melanogaster*) | PBac{erm-flag4C(g)} VK33 | *Janssens and Lee, 2014* doi: 10.1242/dev. 106534. | | |
| Genetic reagent (*D. melanogaster*) | $D^{RNAi}$: $y^1$ $v^1$; P{TRiP. JF02115}attP2 | Bloomington *Drosophila* Stock Center | BDSC: 26217 FlyBase: FBst0026217; RRID:BDSC_26217 | FlyBase symbol: $y^1$ $v^1$; P{TRiP.JF02115}attP2 |
| Genetic reagent (*D. melanogaster*) | $Ase^{RNAi}$: $y^1$ sc* $v^1$ $sev^{21}$; P{TRiP.HMS02847} attP2 | Bloomington *Drosophila* Stock Center | BDSC: 44552 FlyBase: FBst0044552; RRID:BDSC_44552 | FlyBase symbol: $y^1$ sc* $v^1$ $sev^{21}$; P{TRiP.HMS02847}attP2 |
| Genetic reagent (*D. melanogaster*) | $ham^{RNAi}$: $y^1$ $v^1$; P{TRiP.JF02270}attP2 | Bloomington *Drosophila* Stock Center | BDSC: 26728 FlyBase: FBst0026728; RRID:BDSC_26728 | FlyBase symbol: $y^1$ $v^1$; P{TRiP.JF02270}attP2 |
| Genetic reagent (*D. melanogaster*) | $ham^{RNAi}$: $y^1$ sc* $v^1$ $sev^{21}$; P{y[+t7.7] v[+t1.8] =TRiP.HMS00470} attP2 | Bloomington *Drosophila* Stock Center | BDSC: 32470 FlyBase: FBst0032470; RRID:BDSC_32470 | FlyBase symbol: $y^1$ sc* $v^1$ $sev^{21}$; P{y[+t7.7] v[+t1.8]=TRiP.HMS00470} attP2 |
| Genetic reagent (*D. melanogaster*) | $Opa^{RNAi}$: $y^1$ sc* $v^1$ $sev^{21}$; P{TRiP.HMS01185} attP2/TM3, $Sb^1$ | Bloomington *Drosophila* Stock Center | BDSC: 34706 FlyBase: FBst0034706; RRID:BDSC_34706 | FlyBase symbol: $y^1$ sc* $v^1$ $sev^{21}$; P{TRiP.HMS01185}attP2/ TM3, $Sb^1$ |
| Genetic reagent (*D. melanogaster*) | $w^{1118}$; Df(2L) Exel7071/CyO | Bloomington *Drosophila* Stock Center | BDSC: 7843 FlyBase: FBst0007843; RRID:BDSC_7843 | FlyBase symbol: $w^{1118}$; Df(2L)Exel7071/CyO |
| Genetic reagent (*D. melanogaster*) | $ham^{SKI}$, FRT40A/CyO | This paper | | A new hamlet mutant fly line |
| Genetic reagent (*D. melanogaster*) | P{w[+mW.hs]=GawB} elav[C155], P{w[+mC]=UAS-mCD8::GFP.L}Ptp4E [LL4], P{ry[+t7.2]=hsFLP}1, w[*] | Bloomington *Drosophila* Stock Center | BDSC: 5146 FlyBase: FBst0005146; RRID:BDSC_5146 | FlyBase symbol: P{w[+mW.hs]=GawB}elav [C155], P{w[+mC]=UAS-mCD8:: GFP.L}Ptp4E[LL4], P{ry[+t7.2]=hsFLP}1, w[*] |
| Genetic reagent (*D. melanogaster*) | $w^{1118}$; P{w[+mC] =UAS-Dcr-2.D}2 | Bloomington *Drosophila* Stock Center | BDSC: 24650 FlyBase: FBst00024650; RRID:BDSC_24650 | FlyBase symbol: $w^{1118}$; P{w[+mC]=UAS-Dcr-2.D}2 |
| Genetic reagent (*D. melanogaster*) | w*; P{w[+mC]=tubP-$GAL8^{ts}$}2/TM2 | Bloomington *Drosophila* Stock Center | BDSC: 7017 FlyBase: FBst00024650; RRID:BDSC_7017 | FlyBase symbol: w*; P{w[+mC]=tubP- $GAL8^{ts}$}2/ TM2 |
| Genetic reagent (*D. melanogaster*) | $ham^1$,FRT40A/Cyo | *Moore et al., 2002* doi: 10.1126/ science.1072387. | | |
| Genetic reagent (*D. melanogaster*) | UAS-ham | *Moore et al., 2002* doi: 10.1126/ science.1072387. | | |

*Continued on next page*

*Appendix 1—key resources table continued*

| Reagent type (species) or resource | Designation | Source or reference | Identifiers | Additional information |
|---|---|---|---|---|
| Genetic reagent (*D. melanogaster*) | RNAi of *Notch*: $y^1$, $v^1$; $P\{y+t7.7v+t1.8=TRiP.HMS00001\}attP2$ | Bloomington *Drosophila* Stock Center | BDSC: 33611 FlyBase: FBst0033611; RRID:BDSC_33611 | FlyBase symbol: $y^1$, $v^1$; $P\{y+t7.7v+t1.8=TRiP.HMS00001\}attP2$ |
| Genetic reagent (*D. melanogaster*) | *Oregon-R-C* | Bloomington *Drosophila* Stock Center | BDSC: 5 FlyBase: FBst0000005; RRID:BDSC_5 | FlyBase symbol: *Oregon-R-C* |
| Genetic reagent (*D. melanogaster*) | $P\{hsFLP\}^1$, $y^1$ $w^*$; $P\{UAS\text{-}N.intra.GS\}2/CyO$; $MKRS/TM2$ | Bloomington *Drosophila* Stock Center | BDSC: 52008 FlyBase: FBst0052008; RRID:BDSC_52008 | FlyBase symbol: $P\{hsFLP\}^1$, $y^1$ $w^*$; $P\{UAS\text{-}N.intra.GS\}2/CyO$; $MKRS/TM2$ |
| Genetic reagent (*D. melanogaster*) | $y^1$; $M\{vas\text{-}int.Dm\}ZH\text{-}2A$ $w^*$; $M\{UAS\text{-}ham^{\Delta C\text{-}ZF}\text{-}myc\}ZH\text{-}86Fb$ | This paper | | Transgene expressing Hamlet mutant form of the C-terminal zinc finger deletion version |
| Genetic reagent (*D. melanogaster*) | $y^1$; $M\{vas\text{-}int.Dm\}ZH\text{-}2A$ $w^*$; $M\{UAS\text{-}ERD::ham^{N\text{-}ZF}\text{-}myc\}ZH\text{-}86Fb$ | This paper | | Transgene expressing Hamlet the N-terminal zinc finger fused with ERD transcriptional repression domain |
| Genetic reagent (*D. melanogaster*) | $y^1$; $M\{vas\text{-}int.Dm\}ZH\text{-}2A$ $w^*$; $M\{UAS\text{-}VP\text{-}16::ham^{N\text{-}ZF}\text{-}myc\}ZH\text{-}86Fb$ | This paper | | Transgene expressing Hamlet the N-terminal zinc finger fused with VP16 transcriptional activatoin domain |
| Genetic reagent (*D. melanogaster*) | $cu^1$, $tll^{49}/TM3$, $P\{ftz/lacC\}SC^1$, $Sb^1$, $Ser^1$ | Bloomington *Drosophila* Stock Center | BDSC: 7093 FlyBase: FBst007093; RRID:BDSC_7093 | FlyBase symbol: $cu^1$, $tll^{49}/TM3$, $P\{ftz/lacC\}SC^1$, $Sb^1$, $Ser^1$ |
| Genetic reagent (*D. melanogaster*) | $st^1$ $e^1$ $tll^1/TM3$, $Sb^1$ | Bloomington *Drosophila* Stock Center | BDSC: 2729 FlyBase: FBst002729; RRID:BDSC_2729 | FlyBase symbol: $st^1$ $e^1$ $tll^1/TM3$, $Sb^1$ |
| Genetic reagent (*D. melanogaster*) | $ham^{SK4}$, $FRT40A/CyO$ | Bloomington *Drosophila* Stock Center | BDSC: 34329 FlyBase: FBst0034329; RRID:BDSC_34329 | FlyBase symbol: $y^1$ $sc^*$ $v^1$ $sev^{21}$; $P\{TRiP.HMS01316\}attP2$ |
| Genetic reagent (*D. melanogaster*) | $hdac3^{RNAi}$: $y^1$ $sc^*$ $v^1$ $sev^{21}$; $P\{TRiP.HMS00087\}attP2$ | Bloomington *Drosophila* Stock Center | BDSC: 34778 FlyBase: FBst0034778; RRID:BDSC_34778 | FlyBase symbol: $hdac3^{RNAi}$: $y^1$ $sc^*$ $v^1$ $sev^{21}$; $P\{TRiP.HMS00087\}attP2$ |
| Genetic reagent (*D. melanogaster*) | $Su(z)12^{RNAi}$: $y^1$ $sc^*$ $v^1$ $sev^{21}$; $P\{TRiP.HMS00280\}attP2/TM3$, $Sb^1$ | Bloomington *Drosophila* Stock Center | BDSC: 33402 FlyBase: FBst0033402; RRID:BDSC_33402 | FlyBase symbol: $y^1$ $sc^*$ $v^1$ $sev^{21}$; $P\{TRiP.HMS00280\}attP2/TM3$, $Sb^1$ |
| Genetic reagent (*D. melanogaster*) | $Su(var)3\text{-}3^{RNAi}$: $y^1$ $sc^*$ $v^1$ $sev^{21}$; $P\{TRiP.HMS00638\}attP2$ | Bloomington *Drosophila* Stock Center | BDSC: 32853 FlyBase: FBst0032853; RRID:BDSC_32853 | FlyBase symbol: $y^1$ $sc^*$ $v^1$ $sev^{21}$; $P\{TRiP.HMS00638\}attP2$ |
| Genetic reagent (*D. melanogaster*) | $Su(var)205^{RNAi}$: $y^1$ $sc^*$ $v^1$ $sev^{21}$; $P\{TRiP.GL00531\}attP40$ | Bloomington *Drosophila* Stock Center | BDSC: 36792 FlyBase: FBti0146447; RRID:BDSC_36792 | FlyBase symbol: $y^1$ $sc^*$ $v^1$ $sev^{21}$; $P\{TRiP.GL00531\}attP40$ |

*Appendix 1—key resources table continued*

| Reagent type (species) or resource | Designation | Source or reference | Identifiers | Additional information |
|---|---|---|---|---|
| Genetic reagent (*D. melanogaster*) | *UAS-Mi-2$^{DN}$* | **Kovač et al., 2018** doi: 10.1038/s41467-018-04503-2. | | |
| Genetic reagent (*D. melanogaster*) | *UAS-Mi-brm$^{DN}$* | **Herr et al., 2010** doi: 10.1016/j.ydbio.2010.04.006. | | |
| Genetic reagent (*D. melanogaster*) | *In(1)wm4; Su(var)3–91/ TM3, Sb$^1$ Ser$^1$* | Bloomington *Drosophila* Stock Center | BDSC: 6209 FlyBase: FBst0006209; RRID:BDSC_6209 | FlyBase symbol: *In(1)wm4; Su(var)3–91/TM3, Sb$^1$ Ser$^1$* |
| Genetic reagent (*D. melanogaster*) | *w$^{1118}$; PBac {Sp1- EGFP.S} VK00033* | Bloomington *Drosophila* Stock Center | BDSC: 38669 FlyBase: FBst0038669; RRID:BDSC_38669 | FlyBase symbol: *w$^{1118}$; PBac{Sp1- EGFP.S} VK00033* |
| Sequenced-based reagent | Ham_F | **Eroglu et al., 2014** doi: 10.1016/j.cell.2014.01.053. | PCR primers | atagatcctttggccagcagac |
| Sequenced-based reagent | Ham_R | **Eroglu et al., 2014** doi: 10.1016/j.cell.2014.01.053. | PCR primers | agtactcctccctttcggcaat |
| Sequenced-based reagent | Ase_F | **Komori et al., 2014b** doi: 10.7554/eLife.03502. | PCR primers | agcccgtgagcttctacgac |
| Sequenced-based reagent | Ase_R | **Komori et al., 2014b** doi: 10.7554/eLife.03502. | PCR primers | gcatcgatcatgctctcgtc |
| Sequenced-based reagent | D_F | This paper | PCR primers | gcggcggcggtcaacaat |
| Sequenced-based reagent | D_R | This paper | PCR primers | tgcggcgtacagcgaagggt |
| Sequenced-based reagent | Erm_F | **Eroglu et al., 2014** doi: 10.1016/j.cell.2014.01.053. | PCR primers | gttacggccaggcatcgggtcaa |
| Sequenced-based reagent | Erm_R | **Eroglu et al., 2014** doi: 10.1016/j.cell.2014.01.053. | PCR primers | gggccaggcgggattactcgtctc |
| Sequenced-based reagent | PntP1_F | **Komori et al., 2014b** doi: 10.7554/eLife.03502. | PCR primers | ggcagtacgggcagcaccac |
| Sequenced-based reagent | PntP1_R | **Komori et al., 2014b** doi: 10.7554/eLife.03502. | PCR primers | ctcaacgcccccaccagatt |

*Continued on next page*

*Appendix 1—key resources table continued*

| Reagent type (species) or resource | Designation | Source or reference | Identifiers | Additional information |
|---|---|---|---|---|
| Sequenced-based reagent | Dpn_F | *Komori et al., 2014b* doi: 10.7554/eLife. 03502. *Komori et al., 2014b* | PCR primers | catcatgccgaacacaggtt |
| Sequenced-based reagent | Dpn_R | *Komori et al., 2014b* | PCR primers | gaagattggccggaactgag |
| Recombinant DNA reagent | *pUAST-ham$^{\Delta C\text{-}ZF}$-myc-attB (plasmid)* | This paper | | Plasmid DNA of a transgene expressing Hamlet mutant form of the C-terminal zinc finger deletion version |
| Recombinant DNA reagent | *pUAST-ERD::ham$^{N\text{-}ZF}$-myc-attB (plasmid)* | This paper | | Plasmid DNA of a transgene expressing Hamlet the N-terminal zinc finger fused with ERD transcriptional repression domain |
| Recombinant DNA reagent | *pUAST-VP16::ham$^{N\text{-}ZF}$-myc-attB (plasmid)* | This paper | | Plasmid DNA of a transgene expressing Hamlet the N-terminal zinc finger fused with VP16 transcriptional activatoin domain |
| Software, algorithm | LAS AF | Leica Microsystems | RRID:SCR_013673 | |
| Software, algorithm | ImageJ 1.50 g | National Institute of Health | RRID:SCR_003070 | |

