## [Decision Letter]

**Acceptance summary:**

Your demonstration that Tailless is a master regulator of type II neuroblast identity is an important concept and this confirms recent work by Hakes and Brand, 2020. The demonstration that overexpression of Tailless in INPs causes dedifferentiation to type II neuroblasts, while in type I neuroblasts, it causes ectopic type II neuroblasts is of importance. The genetic manipulations performed in the paper showed that Hamlet and Earmuff in the INPs inhibit Tailless, an effect that appears to be mediated by HDac3,

This work therefore brings new sights into how type II neuroblast lineages are specified.

**Decision letter after peer review:**

Thank you for submitting your article "Inactivation of stem cell functionality genes by histone deacetylation enables robust progenitor commitment" for consideration by *eLife*. Your article has been reviewed by Utpal Banerjee as the Senior Editor, a Reviewing Editor, and three reviewers. The reviewers have opted to remain anonymous.

The reviewers have extensively discussed the reviews with one another and the Reviewing Editor has drafted this decision to help you prepare a revised submission.

Summary:

Your demonstration that tailless is a master regulator of type II identity is an important new concept, although this was also recently shown by Hakes and Brand, 2020. It was also important to show that overexpression of Tailless in INPs causes dedifferentiation to type II neuroblasts, while in type I neuroblasts, it causes ectopic type II neuroblasts. Your genetic manipulations allowed you to show that Hamlet and Earmuff in the INPs inhibit Tailless. They also appreciated that the function of Erm and Ham is modified by HDac3, although the reviewers thought that too much attention was given to this effect and suggested to remove it from the title of the paper as of less significance. The reviewers found the work to be novel and convincingly, bringing new sights into how type II neuroblast lineages are specified.

Below is a series of experiments that would correct the weaknesses of the manuscript as well as a very long series of editorial changes that should be very helpful for you to fix small problems.

I should add that the absence of a Materials and methods section was quite disturbing and should not have happened.

Essential revisions:

Below is a series of suggested revisions:

– To confirm if ectopic type II neuroblasts generated by Tll overexpression generate INPs, clones should be done/followed (Figure 2L). Or use the NBGal4 that is expressed in type I NBs to drive UAS-Tll, and focus on a brain region where there is normally no type II NBs, and quantify how many ectopic type II NBs are generating erm-V5+ INPs. A control should be provided to show that there is normally no erm-v5 expression in this region.

– Figure 2C and 2D would benefit from single split channel images, type II lineage should be labelled with a RFP or GFP, to see clearly that type II NBs are ectopically expressing Ase, and they are not type I NBs.

– Figure 3F-H should show split images, very difficult to see which cells are Dpn+ Ase- from these pictures. Also, is there an independent validation that can be done to convince us that these cells are really type II NBs? How about using a Tll antibody to definitively mark these cells, this would also strengthen the point that erm and ham inhibits Tll.

– Figure 6 section – the connection between Hdac3 and *tll* should be explored. What happens to the expression of Tll when Hdac3 is manipulated?

– The authors suggests that Ham and Erm inhibits Tailess, is this specifically in the INPs, or everywhere? An over expression experiment involving UAS-Ham and UAS-Erm in type II NBs would clarify this point.

– The Material and methods section were not available for review which made the interpretation of some experiments difficult. The authors must include the Solexa sequencing raw data in a repository publicly accessible.

Reviewer #1:

This manuscript by Rives-Quinto et al., is addressing the question how intermediate progenitors (INPs) generated by *Drosophila* type II neuroblasts(NBs) stably commit to the INP identity and do not revert back to stem cell fate. There are several novel and significant findings reported. First, Tailless was shown to be necessary and sufficient for the type II neuroblast functional identity; second, Hamlet was identified to be a novel regulator of INP commitment after Erm, and it functions as a transcription repressor, and is required to suppress Tll re-activation by Notch signaling in INPs; Third, Ham and Erm may function through HDAC3. These novel findings illustrate robust mechanisms to down-regulate "stemness" in intermediate progenitors, and will be of great interest to readers of *eLife*.

1) The authors showed modest increase in TllGFP in trans-heterozygous mutant of erm and hamlet that is likely corresponding to the modest increase in the number of type II NBs per lobe in such mutants. I also would like to see how Tll expression changes in hamSK1 homozygous mutant brains. I understand the reason for lacking of this data might be the difficulty to put hamsk1 homozygous mutant and tllGFP together, but Tll antibody is available that will make this experiment feasible.

2) It was shown that erm and ham repress Tll expression in INPs, but whether this is direct or indirect is not known. The interaction between ham, erm and Hdac3 are only shown by genetic interactions. It should be noted that the data presented here suggested an attractive model that erm and ham recruit Hdac3 to *tll* locus to silence its expression in INPs, but further experiments will be needed to confirm this model.

Reviewer #2:

In this manuscript the authors explore the mechanism by which stem cells have the competency to generate intermediate progenitors. The authors use *Drosophila* neuroblasts as a lineage model since it nicely allows them to track cell fates and progenitor competency.

The authors used a type II neuroblast tumor model where they then express in a time-controlled manner insb. Insb causes neuroblasts to differentiate into intermediate progenitors. To identify the genes necessary for type II neuroblast functional identity they sequence the tumors+insb at several time points and look for genes that are downregulated in the differentiating tumor. On the other hand, genes that are upregulated during type II neuroblast differentiation allowed the authors to identify candidate genes important for repressing neuroblast functional identity genes in INPs. This was a very nice strategy and allowed the authors to identify tailless (*tll*) as a type II neuroblast identity gene, earmuff (erm) and Hamlet (Ham) as putative repressors of neuroblast genes in INPs.

Since the material and method section was not available for review, there are some experimental details that are not clear and therefore results that are hard to interpret. The original RNA solexa sequencing data should be made available.

The authors show that RNAi of *tll* in type II neuroblasts leads to neuroblast loss. Tll overexpression with NB>, assuming that NB> is a type I neuroblast driver, causes a conversion from type I to type II neuroblasts. It is however not clearly demonstrated that these ectopic type II neuroblasts are capable of generating INPs. The role of Tll in type II neuroblasts was recently described in another manuscript (Hakes and Brand, *eLife*) and this should be discussed in this manuscript.

By a series of epigenetic experiments, the authors find that erm and ham genetically interact and are both required to maintain INP commitment, preventing their reversion to neuroblasts. They further show that reducing the levels of both erm and ham (double heterozygous) is sufficient to de-repress *tll* in INPs.

By using a series of ham mutants and domain analysis the authors claim that the N-terminus zinc finger of Ham is required to repress neuroblast genes in INPs. These experiments and result interpretation are however not clear.

Because *tll* was previously described as a putative target of Notch, the authors explore how Notch regulates INP commitment. The authors show that reducing Notch levels rescues the ham mutant phenotype, and that Notch constitutive activation enhances the phenotype of erm, ham double heterozygous. The connection between Notch and Tll was not experimentally explored.

To test the hypothesis that Erm and Ham required chromatin repressing complexes to repress neuroblast genes in INPs the authors tested the role of several candidate chromatin regulators. The RNAi of Hdac3 in type II neuroblasts, or in the background of either ham or erm heterozygotes causes an increase in type II neuroblast number. Since reducing Hdac3 levels by itself does not completely mimic ham or erm mutant phenotypes, this might indicate that Hdac3 is not the only chromatin regulator acting downstream of these genes.

Overall, the authors characterize the function of several genes responsible for either neuroblast or progenitor fate, but the connection between *tll* and Notch or *tll* and erm/Ham is not as clear.

– The Material and methods section were not available for review which made the interpretation of some experiments difficult. The authors should include the Solexa sequencing raw data in a repository publicly accessible.

– The title does not represent what has been clearly shown in this manuscript and should be altered. The authors should not emphasize the chromatin regulation aspect.

– Abstract should be modified as well according to new title.

– Tll::GFP expression pattern in type 1 neuroblasts should be shown. This is essential to corroborate that Tll is a type II neuroblast specific functional gene (subsection “A novel transient over-expression strategy to identify regulators of type II neuroblast functional identity”). Please show an image of an entire central brain lobe (dorsal and ventral) and an image of a ventral nerve cord (related to Figure 1).

– To confirm if ectopic type II neuroblasts generated by Tll overexpression generate INPs, clones should be done/followed (Figure 2L).

– Subsection “Ham is a new regulator of INP commitment” – the authors need to say what are the candidate transcription factors identified. Could be included in a table and text. The transition from analysing *tll* to analyzing erm needs to be re-written as it is hard to follow.

– Every time erm, ham double heterozygotes are used the exact alleles used should be included in the text and figures (e.g. subsection “Ham promotes stable INP commitment by repressing gene transcription”).

– All figures – uniformize genotypes shown in graphs and labels. Homozygous mutants should either be represented as e.g. Ham sk1/sk1 or HamSk1 -/-.

– All figures should say the stages being analyzed.

– Figure 3 – legend is very incomplete. Legend should be reviewed and information about the Ham alleles provided. It is not clear what the SK mutants really are molecularly.

– In the legend of Figure 3 it is mentioned that Ham is expressed in iINPs and INPs. This is hard to see in the figure which is small. Is Ham expressed in the NB? It would be helpful to know to distinguish role in INP formation vs. commitment.

– In Figure 3C – since the information about the Ham antibody is not available, it is not clear if the antibody should recognize HamSk1 and SK4 since the majority of the protein should be produced. Assuming the authors used the previously published Ham Ab, which recognizes the N-terminus of Ham, the authors should discuss why there is no staining in SK1 mutants.

– Section regarding Figure 4 (subsection “Ham promotes stable INP commitment by repressing gene transcription”) – It is concluded that the N-terminus ZF domain of Ham is the domain responsible for repressing neuroblast genes in INP. However, in both Ham1 and in HamSK1, the N-ZF domain are present and wild-type. Since in HamSk, where this N-ZF is present there is a phenotype, how can it be concluded that this is the domain responsible? Unless this SK mutant causes Ham degradation and, in the end, re-expressing Ham, either full-length or just the functional domain rescues the KO phenotype. The authors should clarify this experimentally and in the text.

– Figure 5 – 5C include *tll*::GFP in wt background.

– Subsection “Inactivation during INP commitment relinquishes Notch’s ability to activate *tll* in INPs” – rephrase this sentence. The connection between *tll* and Notch is not demonstrated, pose as an hypothesis.

– Subsection “Inactivation by Hdac3 relinquishes Notch’s ability to activate *tll* in INPs” – rephrase to say that "Ham overexpression can replace endogenous Erm function". In the wild-type situation this is not true as both erm or ham mutants have a phenotype.

– Figure 6 section – the connection between Hdac3 and *tll* should be explored. What happens to the expression of Tll when Hdac3 is manipulated?

– Figure 7 – why is *tll* off in the erm or ham loss of function? This was not shown, and in Figure 5C and D, double heterozygotes erm, ham cause *tll* expression in INPs, the opposite phenotype of what is in the model.

The model should be re-drawn as it is hard to interpret and contains too many speculations.

Reviewer #3:

Lee and Colleagues have previously shown that the nuclear protein Insensible inhibits the activity of notch downstream proteins such as dpn through culling mediated proteolysis and Insb over expression triggers premature differentiation in type II NBs. Utilising this tool, they show that in Brat mutant brains, over expression of Insb can promote differentiation, and performing RNAseq, they found that tailess is a master regulator of type II identity. Consistent with what was recently shown by Hakes and Brand, 2020, Tailess over expression in INPs cause dedifferentiation to type II NBs, and its over expression in type I NBs causes ectopic formation of type II NBs. Using some very nice genetics, the authors show that Hamlet and Erm in the INPs are required to inhibit Tailess. Furthermore, Tailess has previously been shown to be activated by Notch. In the INPs, they show that Notch intra is less capable of inducing INP to Type II NB reversion in the absence of Erm and Ham. Finally, the repressive function of Erm and Ham is modified by HDac3. The work is novel, and convincingly shows that ham and erm are required for Tll inactivation, and brings new sights into how type II NB lineages are specified. This work is suitable for publication in *eLife*.

---

## [Author Response]

Essential revisions:Below is a series of suggested revisions:– To confirm if ectopic type II neuroblasts generated by Tll overexpression generate INPs, clones should be done/followed (Figure 2L). Or use the NBGal4 that is expressed in type I NBs to drive UAS-Tll, and focus on a brain region where there is normally no type II NBs, and quantify how many ectopic type II NBs are generating erm-V5+ INPs. A control should be provided to show that there is normally no erm-v5 expression in this region.

We thank reviewers for the suggestion.

We overexpressed *tll* driven by GAL4 drivers of various strength, and found that highly levels of Tll overexpression are required to convert type I neuroblasts in the ventral brain region into type II neuroblasts.

We have added a new panel of images (Figure 2N-P) showing transformed type II neuroblasts in the ventral-medial region of the brain generate immature INPs marked by Erm::V5 expression.

We added quantification of total type I neuroblasts per ventral brain lobe that overexpressed a *UAS-tll* driven by a pan-neuroblast Gal4 (Figure 2K). This result shows that Tll overexpression transforms greater than 60% of ventral type I neuroblasts (Dpn+Sp-1::GFP-Ase+) into type II neuroblasts (Dpn+Sp-1::GFP+Ase-).

We also added a new paragraph (Discussion) to discuss how brain.

– Figure 2C and 2D would benefit from single split channel images, type II lineage should be labelled with a RFP or GFP, to see clearly that type II NBs are ectopically expressing Ase, and they are not type I NBs.

We thank reviewers for the suggestion.

We have now included single-channel, higher-magnification images of brain samples shown in Figure 2C and 2D. We encircled type II neuroblast lineages with yellow dotted lines, and used colored arrows and arrowheads to label distinct cell types in the lineage. A key to colored arrows and arrowheads is provided in the legend for every figure.

Three lines of evidence led us to conclude that *tll*-mutant type II neuroblasts indicated by white arrows ectopically express Ase. First, the *UAS-tllRNAi* transgene was driven by a type II neuroblast-specific Gal4 driver (*Wor-Gal4,Ase-Gal80*), which has been well-characterized and widely used in many studies. Second, *tll*-mutant type II neuroblasts (shown in Figure 2D) are surrounded by immature INPs and INPs, identical to type II neuroblasts in wild-type brains (shown in Figure 2C). Third, the diameter of *tll*-mutant type II neuroblasts is significantly smaller than type I and type II neuroblasts in wild-type brains, and reduced cell diameter is a typical diagnostic marker of neuroblast differentiation. Ectopic Ase expression is only detected in the late stage of type II neuroblast differentiation. One of the type II neuroblasts in *tll* knock-down brains clearly show strong ectopic Ase expression, but the neighboring type II neuroblast has yet to upregulate Ase expression. These reasons led us to conclude that *tll*-mutant type II neuroblasts shown in Figure 2D ectopically express Ase due to premature differentiation.

– Figure 3 F-H should show split images, very difficult to see which cells are Dpn+ Ase- from these pictures. Also, is there an independent validation that can be done to convince us that these cells are really type II NBs? How about using a Tll antibody to definitively mark these cells, this would also strengthen the point that erm and ham inhibits Tll.

We thank reviewers for the suggestion.

We have included single-channel images of brain samples shown in Figure 3F-H to allow for clear visualization of type II neuroblasts (Dpn+Ase-).

We obtained a commercially available Tll antibody used in a previously published study, but this antibody was unable to detect specific signals in larva brains after trying multiple immunofluorescent staining protocols.

– Figure 6 section – the connection between Hdac3 and tll should be explored. What happens to the expression of Tll when Hdac3 is manipulated?

We thank reviewers for the suggestion.

We have now included a panel of images and quantification of Tll::GFP expression relative to Dpn expression in INPs in Figures 6E-G. We have also added a paragraph (subsection “Inactivation by Hdac3 relinquishes Notch’s ability to activate *tll* in INPs”) describing findings from this experiment.

– The authors suggests that Ham and Erm inhibits Tailess, is this specifically in the INPs, or everywhere? An over expression experiment involving UAS-Ham and UAS-Erm in type II NBs would clarify this point.

We thank reviewers for the suggestion.

We have now included a panel of images and quantification of Tll::GFP expression relative to Dpn expression in neuroblasts in Figures 5F-I. We added text (subsection “Inactivation during INP commitment relinquishes Notch’s ability to activate *tll* in INPs”) describing findings from this experiment.

– The Material and methods section were not available for review which made the interpretation of some experiments difficult. The authors must include the Solexa sequencing raw data in a repository publicly accessible.

We apologize for our mistake. We added the Materials and methods section.